# Should I Have Expressed a Different Intent?
# Counterfactual Generation for LLM-Based Autonomous Control

**Amirmohammad Farzaneh** [1]   **Salvatore D'Oro** [2]   **Osvaldo Simeone** [1]

## Abstract

Large language model (LLM)-powered agents can translate high-level user intents into plans and actions in an environment. Yet after observing an outcome, users may wonder: What if I had phrased my intent differently? We introduce a framework that enables such counterfactual reasoning in agentic LLM-driven control scenarios, while providing formal reliability guarantees. Our approach models the closed-loop interaction between a user, an LLM-based agent, and an environment as a structural causal model (SCM), and leverages test-time scaling to generate multiple candidate counterfactual outcomes via probabilistic abduction. Through an offline calibration phase, the proposed conformal counterfactual generation (CCG) yields sets of counterfactual outcomes that are guaranteed to contain the true counterfactual outcome with high probability. We showcase the performance of CCG on a wireless network control use case, demonstrating significant advantages compared to naive re-execution baselines.

## 1. Introduction

**Context and Motivation:**   Large language models (LLMs) are increasingly being used as decision-making agents in autonomous control pipelines (Huang et al., 2022; Yao et al., 2023). In such agentic AI systems, a human user specifies a high-level goal or *intent* in natural language, and an LLM-based agent must interpret this intent, plan a sequence of actions, execute them, observe the outcomes, and finally report back to the user.

A recent example of such settings is given by AgentRAN

[1] Institute for Intelligent Networked Systems, Northeastern University, London, UK [2] Institute for Intelligent Networked Systems, Northeastern University, Boston, MA, USA. Correspondence to: Amirmohammad Farzaneh <a.farzaneh@nulondon.ac.uk>.

*Proceedings of the 43$^{rd}$ International Conference on Machine Learning*, Seoul, South Korea. PMLR 306, 2026. Copyright 2026 by the author(s).

(Elkael et al., 2025). AgentRAN is an agentic network control framework in which, as illustrated in Fig. 1, a wireless cellular network operator provides a plain-text intent, such as a configuration request. The LLM agent issues corresponding network commands, and the network returns *key performance indicators* (KPIs). Finally, the agent generates a summary report of the outcome for the operator.

Once the agent presents its report, the human may naturally ask "what if" questions about alternative instructions or intents. For instance, in the AgentRAN scenario, the operator may wonder what would have happened if the intent had been phrased differently or if different KPI targets had been indicated to the LLM-based agent. An example can be found in Fig. 2.

Addressing such counterfactual queries requires reasoning about *both* the agent's internal decision process and the environment's response under the hypothetical counterfactual scenario that a different intent had been expressed. All else being equal, a different user prompt could cause the agent to choose a different plan of actions, which in turn would induce a different environment trajectory and outcome. While the agent's internal operation is typically accessible, making it possible to study counterfactual actions by the agent (Chatzi et al., 2025; Ravfogel et al., 2025), reasoning about alternative environment responses requires a *simulation-in-the-loop* approach (Hu et al., 2025). Developing a method to reliably evaluate these counterfactual outcomes – as if the user had expressed a different intent — is the problem we address in this work.

**Main Related Work:**   **Counterfactual generation** with LLMs has been recently studied in text-generation settings by reformulating LLMs as structural causal models (SCMs) (Pearl, 2009; Peters et al., 2017). This enables the generation of *true* counterfactual text by reusing the same sampling noise for original and perturbed prompts (Chatzi et al., 2025; Ravfogel et al., 2025). However, these methods do not account for interactions with an external environment.

The **counterfactual inference** of network KPIs under alternative configurations was studied in (Hou et al., 2025), building on the framework in (Lei & Candès, 2021). These works apply to settings in which the operator directly sets ac-

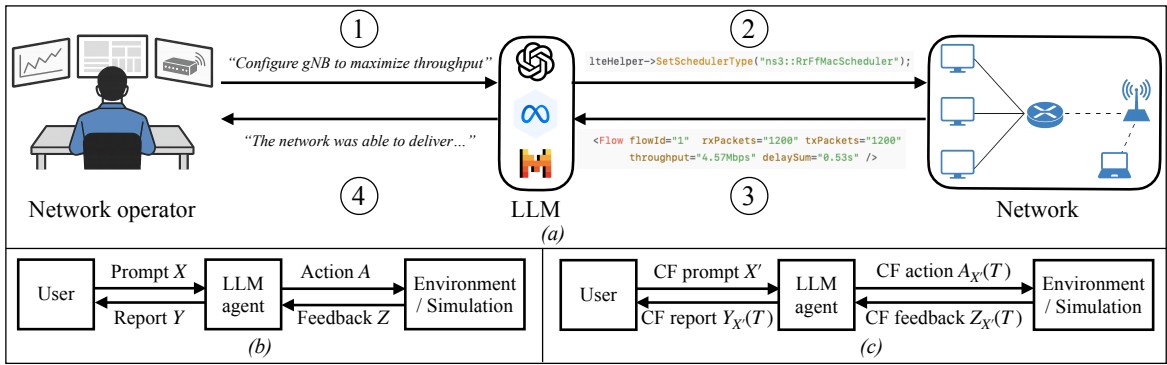

*Figure 1.* (a) Example of the agentic framework studied in this work: A network operator provides an intent as a prompt to an LLM-based agent. The agent translates this prompt into an action, which is used to configure the network. The network executes the action, and returns environment feedback to the LLM in the form of KPIs, which the LLM processes and summarizes into a report for the operator. (b) LLM-environment interaction model: The user provides a prompt $X$, which the LLM maps to an action $A$. The environment executes action $A$ and produces feedback $Z$. The LLM then combines $(X, A, Z)$ to generate the response $Y$, which is returned to the user. (c) Given a factual episode $T = (X, A, Z, Y)$, the network operator chooses a counterfactual prompt $X'$. Given the observed factual episode $T$, our goal is to estimate the counterfactual report $Y_{X'}(T)$ that the user would have received, under the same conditions, had the input prompt been $X' \in \mathcal{E}(X)$ instead of $X$ (see Fig. 2 for an example).

| Factual intent ($X$) | True counterfactual report (Y_X') |
|---|---|
| Context: 5 UEs with **time-varying** SNRs. Each UE has expected offered load 1.20 Mbps. Goal: Configure gNB to **maximize aggregate throughput** while promoting fairness across UEs. Targets: Per-UE throughput around **1.08 Mbps** on average; average per-UE latency $\leq$ **18 ms.** Summarize the KPIs over a period of 10 seconds. | Context: 5 UEs with **static** SNRs. Each UE has expected offered load **1.20 Mbps**. Goal: Configure gNB to **maximize fairness** and latency stability. Targets: Per-UE throughput around **0.84 Mbps** on average; average per-UE latency $\leq$ 10 ms. Summarize the KPIs over a period of 10 seconds. |

| Factual report ($Y$) | True counterfactual report ($Y_{X'}$) |
|---|---|
| Chosen scheduler: PF. Over 10 s, the system achieves an average throughput of 1.07 Mbps per UE, with total throughput around 5.3–5.5 Mbps for most of the run. The average end-to-end latency remains between 15–18 ms. A brief dip occurs at $t \approx 4.4$ s, where total throughput drops to about 5.0 Mbps before recovering quickly. | Chosen scheduler: RR. The system delivers stable throughput at approximately 0.84 Mbps per UE (total $\approx$4.2 Mbps), with very small spread across users. The average latency stays tightly within 8–10 ms. A short dip is observed near $t \approx 4.5$s, where total throughput decreases to about 3.9 Mbps, followed by a smooth return to the previous level. |

| Interventional generation (IG)-based report ($\hat{Y}_{X'}$) | Counterfactual generation (CG)-based report ($\hat{Y}_{X'}(T)$) |
|---|---|
| Chosen scheduler: RR. In this run, total throughput varies between 3.4–5.0 Mbps, with per-UE rates ranging from 0.65–1.05 Mbps. Latency fluctuates between 7–14 ms, with several intervals above the 10 ms target. A pronounced dip occurs around $t \approx 2.5$ s, followed by a prolonged low-throughput period after $t \approx 6$ s. | Chosen scheduler: RR. Per-UE throughput remains close to 0.84 Mbps (total $\approx$4.2 Mbps) with uniform allocation. Average latency stays low (8–9 ms). A mild dip appears near $t \approx 4.4$ s before fast recovery. |

*Figure 2.* An example of a factual episode with a true and a generated counterfactual report. Emphasis (bold fonts) added to highlight the main differences between factual and counterfactual intents.

tions and observes outcomes. Thus, they do not account for the presence of an LLM agent as the intermediary between operator and system.

Recent work on **conformal language modeling** (CLM) has introduced a principled framework for endowing LLM text generation with statistical guarantees. This is done by producing sets of candidate generations whose miscoverage with respect to a task-specific admission function is controlled via conformal calibration (Quach et al., 2024). See also (Angelopoulos et al., 2023; Geifman & El-Yaniv, 2017; Jiang et al., 2020) for related work. All these prior studies focus on standalone text generation and do not account for closed-loop interactions between an LLM agent and an external environment.

**Main Contributions:** This work addresses the problem of reliable counterfactual generation in LLM-powered autonomous control systems, enabling operators to obtain trustworthy answers to the question "What would have happened if I had expressed a different intent?" As exemplified

in Fig. 2, counterfactual prompts may capture the following types of differences with respect to the factual prompt: (*i*) different environmental contexts; (*ii*) different design goals and targets of the user; and (*iii*) different prompt designs expressing the same contexts, goals, and target (i.e., paraphrases of the original intent).

● **Agent-environment counterfactual generation:** We introduce an SCM, shown in Fig. 3, that unifies token-level LLM counterfactual reasoning (Chatzi et al., 2025; Ravfogel et al., 2025) with environment-level causal inference (Peters et al., 2017). The SCM incorporates the true LLM architecture for the agent, and a learned simulator for the environment. Based on the SCM in Fig. 3, we introduce a **counterfactual generation** (CG) methodology that proceeds via abduction to produce an estimate of the true counterfactual report in response to a counterfactual prompt.

● **Conformal counterfactual generation:** Through testtime scaling, CG can produce a set of counterfactual reports. We introduce **conformal CG** (CCG), a calibration

*Figure 3.* SCM of the agent-environment pipeline: The user prompt $X$ influences the action $A$ chosen by the LLM **action generator**, which in turn affects the environment feedback $Z$. Finally, the LLM **report generator** combines $(X, A, Z)$ to generate the response $Y$. The exogenous variables $U_A$, $U_Z$, and $U_Y$ capture the randomness specific to the action generator, the environment, and the report generator, respectively, and are assumed independent of $X$ and of each other.

framework that ensures that, with high probability, at least one element of the set constitutes a semantically faithful approximation of the true counterfactual outcome, while minimizing the cardinality of the set (see Fig. 4).

• **Experimental results:** Through experiments in a network control scenario and a drone navigation task (Appendix I), CG is verified to produce accurate and stable counterfactual reports, outperforming naive re-execution baselines. Furthermore, CCG is seen to achieve coverage guarantees, while substantially reducing the number of excess counterfactual samples relative to fixed-budget baselines. For example, in our 5G network control case study, CCG provides operators with a small set of plausible counterfactual reports that is both faithful to the true counterfactual outcome and accompanied by an explicit reliability guarantee.

## 2. Problem Definition

As illustrated in Fig. 1, we consider an agentic AI workflow in which a user specifies an intent in natural language; an LLM-based agent converts this intent into actions; an external real-world environment executes those actions, returning measurable feedback; and the agent responds to the user with a natural-language report on the outcome of its action on the environment.

We define an episode $T$ as the quadruple $T = (X, A, Z, Y)$, where $X \in \mathcal{X}$ is the **user prompt**, expressing its intent in plain text; $A \in \mathcal{A}$ is the sequence of **actions** issued by the agent and executed in the environment; $Z \in \mathcal{Z}$ is the environment **feedback**, describing the response of the environment to the agent's actions; $Y \in \mathcal{Y}$ is the textual **report** returned to the user as a function of the feedback $Z$. As exemplified in Fig. 2, the prompt $X$ may encode contextual information as described by the user. This information may not represent the true environmental context, which, as detailed in the next section, must be inferred by the system via a simulator-in-the-loop. Extensions of the problem setting may also incorporate explicit variables encoding partial environmental information available at the user.

Given a factual prompt $X$, we define a task-specific set of admissible **counterfactual prompts** $\mathcal{E}(X) \subseteq \mathcal{X}$. The prompts in set $\mathcal{E}(X)$ are obtained from the original prompt $X$ through edits that express changes the user may wish to make in order to analyze the alternative intents or different descriptions of the same intent. In practice, these may also be generated by an LLM based on user's instructions. Each such edit produces a valid counterfactual prompt $X' \in \mathcal{E}(X)$. An example of a counterfactual prompt is shown in Fig. 2.

Given an observed factual episode $T = (X, A, Z, Y)$, our objective is to estimate the counterfactual report $Y_{X'}(T)$ that the user would have received, under the same conditions, had the input prompt been $X' \in \mathcal{E}(X)$ instead of $X$.

In our framework, there are two LLM components with distinct roles. The first component, the **action generator**, maps the natural-language intent $X$ into an executable action representation $A$ that is directly consumable by the environment. To ensure that $A$ is unambiguous and tool-ready, this component is prompted with strict formatting instructions to output a structured object that matches the environment interface. For example, it may output a JSON configuration specifying scheduler type, number of UEs, traffic load, and simulation duration. The second component, the **report generator**, takes as input the triplet $(X, A, Z)$ and produces the textual report $Y$. In practice, both components may rely on the same underlying base model, but are invoked with different system prompts.

## 3. Counterfactual Generation

In this section, we present our methodology for generating point estimates $\hat{Y}_{X'}(T)$ of counterfactual reports. We begin by introducing in Sec. 3.1 an SCM of the agent–environment system, and then describe in Sec. 3.2 how to generate counterfactual actions, environment feedback, and textual reports for edited prompts $X'$. Sec. 3.3 reports numerical results.

### 3.1. Modeling the Agent–Environment System as a Structural Causal Model

To formally reason about counterfactuals in the agent–environment system of Fig. 1, we adopt the framework of SCMs (Peters et al., 2017). An SCM specifies, for each endogenous variable, a functional relationship with a subset of other endogenous variables, along with an associated exogenous noise term. In our setting, the endogenous variables are given by the user prompt $X$, the agent action $A$, the environment feedback $Z$, and the final LLM report $Y$, i.e., by the tuple $(X, A, Z, Y)$. The exogenous noise captures any randomness associated with the generation of the given endogenous variable that is unexplained by the functional relationship with other endogenous variables.

As illustrated in Fig. 3, actions $A$ and report $Y$ are organized into tokens, which are denoted as $A_1, A_2, \ldots$, and $Y_1, Y_2, \ldots$, respectively.

1) **Action generator:** Consider first the generation of the action sequence $A_1, A_2, \ldots$. Denote as $V$ the size of the LLM dictionary. Following (Chatzi et al., 2025), given the input prompt $X$ and the past actions $A_1, \ldots, A_{i-1}$, the new action $A_i$ is generated via a Gumbel–Max mechanism. As argued in (Chatzi et al., 2025), this sampling mechanism ensures that counterfactual sequences stay semantically "close" to the factual generation under small changes in the intent $X$.

(1) **Next-token probability distribution:** The LLM produces the next token distribution $P_{A,i} = [P_{A,i,1}, \ldots, P_{A,i,V}]^T$, where the $v$-th entry $P_{A,i,v}$ represents the probability assigned to the $v$-th element of the vocabulary.

(2) **Next-token sampling via Gumbel-Max:** The exogenous noise $U_{A,i}$ is a $V \times 1$ vector whose entries are sampled independently from the standard Gumbel distribution, i.e.,

$$U_{A,i} = [U_{A,i,1}, \ldots, U_{A,i,V}] \overset{\text{i.i.d.}}{\sim} \text{Gumbel}(0, 1). \quad (1)$$

The next token $A_i$ is then sampled as

$$A_i = \underset{v \in \{1, \ldots, V\}}{\arg \max} \{\log P_{A,i,v} + U_{A,i,v}\}. \quad (2)$$

By the properties of the Gumbel distribution, the Gumbel-Max mechanism (2) ensures that the distribution of the next token $A_i$ is $P_{A,i}$ (Chatzi et al., 2025).

2) **Environment model:** Given actions $A$, the environment produces output $Z$, which is modeled via the functional relationship

$$Z = f_Z(A, U_Z), \quad (3)$$

where $U_Z$ represents exogenous variables, which are independent of all other variables. Unlike the LLM-based action generator, which describes the *true* operation of the agent,

the function (3) is only an *approximation* of the real physical environment. In practice, the function $f_Z(A, U_Z)$ is implemented via a simulation of the environment, such as a digital twin (Hu et al., 2025; Ruah et al., 2023; Testolina et al., 2024), with the exogenous variables $U_Z$ representing a random seed or explicit variables describing environmental context.

3) **Report generator:** As seen in Fig. 3, the report generator follows the same autoregressive and noise-augmented structure as the action generator, but is conditioned on the tuple $(X, A, Z)$ rather than on the prompt $X$ alone. We denote the corresponding collection of exogenous Gumbel variables as $U_Y = (U_{Y,1}, U_{Y,2}, \ldots)$, which are drawn independently from those of the action generator $U_A$.

### 3.2. Counterfactual Generation

Given the SCM in Fig. 3, and given a factual episode $T$, CG proceeds through the following steps:

(1) **Prompt Editing:** Given a factual prompt $X$, the user or system specifies a counterfactual prompt $X' \in \mathcal{E}(X)$ obtained by editing the original intent.

(2) **Abduction:** From the observed factual episode $T$, the system infers the exogenous noise variables $U_A = (U_{A,1}, U_{A,2}, \ldots)$, $U_Y = (U_{Y,1}, U_{Y,2}, \ldots)$, and $U_Z$. The LLM exogenous variables $U_A$ and $U_Y$ are known, as they are generated by the LLM agent in the process of producing the outputs $A$ and $Y$ via Gumbel-Max sampling. In contrast, the environment exogenous variables $U_Z$ are latent, since the true output $Z$ is produced by the environment, and not by the model (3). Therefore, the exogenous variable $U_Z$ must be inferred from the observed trace $T = (X, A, Z, Y)$ through an abduction step to be discussed below (Pearl, 2009), producing an estimate $\hat{U}_Z$.

(3) **Action Counterfactual Generation:** Replacing the factual prompt $X$ with the counterfactual prompt $X'$, the LLM agent's autoregressive decoding process (2) is replayed with $X'$ and $U_A$ to generate the counterfactual action sequence $A_{X'}(T)$, reflecting how the agent would have acted if prompted differently.

(4) **Environment Counterfactual Simulation:** Using the counterfactual action sequence $A_{X'}(T)$ and the inferred environment noise $\hat{U}_Z$, the environment model (3) is run to produce counterfactual feedback $\hat{Z}_{X'}(T) = f_Z(A_{X'}(T), \hat{U}_Z)$.

(5) **Report Counterfactual Generation:** Finally, the report generating LLM is run with inputs $(X', A_{X'}(T), \hat{Z}_{X'}(T))$ and exogenous $U_Y$ to generate the counterfactual report $\hat{Y}_{X'}(T)$.

As discussed, CG requires the inference of the exogenous variables $U_Z$ corresponding to the factual pair $(A, Z)$. To

this end, we start by fixing a prior distribution $p(U_Z)$, which may be uniform on the simulator random seed or may account for standard modeling assumptions (Ruah et al., 2023). Given a factual pair $(A, Z)$, abduction estimates the posterior distribution $p(U_Z|A, Z)$ implied by the SCM in Fig. 3 under the prior $p(U_Z)$.

Given that the model $f_Z(A, U_Z)$ is typically a black-box simulator, standard likelihood-based methods are not applicable. Therefore, we resort to *likelihood-free methods* (Cranmer et al., 2020; Lueckmann et al., 2021; Papamakarios et al., 2019). We specifically adopt the state-of-the-art *neural posterior estimation* (NPE) approach (Greenberg et al., 2019). In this framework, a neural network $q_\phi(U_Z|A, Z)$ with parameters $\phi$ is trained to approximate the intractable posterior $p(U_Z|A, Z)$ using samples generated from the forward model. The estimate $\hat{U}_Z$ is then obtained by sampling as $\hat{U}_Z \sim q_\phi(U_Z \mid A, Z)$. We refer to Appendix E for the details.

### 3.3. Experiments

In this section, we present numerical results on counterfactual generation for agentic AI-based networks.

#### 3.3.1. SETUP AND BENCHMARKS

1) **Setup:** We study a representative networking workflow where a user asks an LLM to configure a 5G base station (gNB) by determining the number of UEs to be served by the gNB, the scheduler, the traffic load requirements, and the observation period; observe the relevant KPIs; and finally summarize the results to the user (see Fig. 1). The set of allowed counterfactual prompts $\mathcal{E}(X)$ supports controlled edits in which one or more gNB configuration parameters are modified. An example of such a counterfactual prompt can be found in Fig. 2. The real-world environment, emulated via ns-3 (ns-3 Consortium, 2023), encompasses a single macro gNB operating in a 5G NR cell with multiple UEs randomly distributed in the coverage area. The emulator supports Round-Robin (RR) and Proportional Fair (PF) schedulers, log-distance path loss with shadowing and Rayleigh fading, and stochastic per-UE traffic arrivals. The resulting KPIs $Z$ consist of time series of per-UE throughputs and end-to-end delays sampled every 0.2 seconds.

The black-box simulator function $f_Z(A, U_Z)$ is a "digital twin" that is also implemented using ns-3 using the same single-cell topology as the real environment, but with simplified channel and traffic models to account for the limited realism of a practical surrogate model. A similar approach was used in (Hu et al., 2025). In particular, the simulator $f_Z(A, U_Z)$ replaces fast fading with an averaged link gain model, and omits packet-level queueing jitter. An ablation study analyzing the impact of simulator fidelity on both counterfactual generation accuracy and conformal sampling

efficiency is reported in Appendix H.

To evaluate counterfactual reasoning, we constructed a dataset comprising 300 original prompts $X$ and their corresponding counterfactual edits $X'$. These prompts were generated automatically using GPT-4 (OpenAI, 2023) which was prompted to produce realistic operator-level configuration requests and their counterfactual variations (see details in Appendix B). For each episode, we recorded both the factual response $Y$ produced under the original prompt $X$ and the true counterfactual response $Y_{X'}(T)$ obtained by rerunning the real-world emulator under the counterfactual prompt $X'$ while keeping the same random seed. We employ Llama 3 8B-Instruct (Touvron et al., 2023) as the LLM agent.

2) **Benchmarks:** We compare the performance of the proposed CG methods obtained with two alternatives:

1. **Interventional Generation (IG):** In the IG method, the estimated counterfactual report $\hat{Y}_{X'}$ is obtained by feeding the counterfactual prompt $X'$ directly to the agent-environment system. This approach neglects the factual episode $T$, implementing an independent run of the entire system under the new prompt $X'$. Note that, unlike CG, this benchmark requires a separate use of the real system to produce the estimate $\hat{Y}_{X'}$.

2. **Simulated Interventional Generation (SIG):** SIG operates as IG with the only difference that the counterfactual report $\hat{Y}_{X'}$ is produced by connecting the agent to the simulator $f_Z(A, U_Z)$, where the random seed $U_Z$ is drawn from the prior. Like IG, SIG neglects the factual episode $T$; and, like CG, SIG does not require access to the real environment.

#### 3.3.2. RESULTS

We begin by studying the quality of the counterfactual KPI outcomes $\hat{Z}_{X'}(T)$ produced by the agent-environment system for an episode $T$ and counterfactual query $X'$ using IG, SIG, and CG. To this end, we assess how closely the KPI sequences $\hat{Z}_{X'}(T)$ produced by IG, SIG, and CG track the true counterfactual signal $Z_{X'}(T)$. We evaluate the **mean absolute error** (MAE), the maximum **cross-correlation** after allowing up to a 10 ms lead/lag, and the **crossing-level error**, i.e., the absolute difference in the fraction of time spent above thresholds 5 Mbps and 15 ms for throughput and latency, respectively.

Table 1 summarizes the results of our experiments over 100 test examples. Across all metrics, CG consistently tracks the true counterfactual KPIs more closely than IG and SIG, reflecting higher fidelity in both pointwise and temporal correlation structure. This is despite the fact that IG requires access to the real system for counterfactual estimation.

*Table 1.* Fidelity in reconstructing the counterfactual KPI time series (best in **bold**).

| Metric | CG / IG / SIG | | |
| --- | --- | --- | --- |
| | MAE | Cross-corr. peak | Crossing-level error |
| Throughput | **0.15** / 0.28 / 0.33 | **0.93** / 0.78 / 0.70 | **0.03** / 0.14 / 0.18 |
| Delay | **0.35** / 0.52 / 0.60 | **0.89** / 0.71 / 0.62 | **0.05** / 0.20 / 0.26 |

We now turn to evaluating the semantic quality of the counterfactual reports $\hat{Y}_{X'}(T)$ generated using IG, SIG, and CG. In this experiment, the AI agent is tasked with producing a descriptive analysis of the time series obtained from the environment. Fig. 2 illustrates an example for one test point, showing the true counterfactual report alongside the estimated counterfactual reports produced by IG and CG. SIG produced a counterfactual report similar to IG, and it is not shown in the figure.

To systematically compare true and estimated reports, we adopt an LLM-as-a-judge framework (Gu et al., 2024). For each test case, the judge LLM (GPT-4) is provided with the true counterfactual report as a reference and three candidate reports generated by IG, SIG, and CG, and is asked to select the candidate that most closely matches the reference in content and semantics (see Appendix C). Across 100 test cases, CG is selected as the best match in 92% of cases, while IG is preferred in the remaining 8%, and SIG is never selected. This result indicates a clear advantage of CG over the alternatives, and the weaker performance of SIG highlights the additional semantic distortion introduced by the simulator.

# 4. Conformal Calibration of Counterfactual Responses

In the previous section, we described CG, a methodology capable of producing point estimates $\hat{Y}_{X'}(T)$ of the true counterfactual report $Y_{X'}(T)$. Following the principle of test-time scaling (Quach et al., 2024), the CG pipeline can be run multiple times to produce a set of counterfactual reports $\hat{Y}_{X'}(T)$. In this section, as illustrated in Fig. 4, we aim to calibrate the criteria used to determine whether to accept each generated report and to stop generating. Calibration aims at ensuring the quality and diversity of the obtained set of generated counterfactual reports with respect to the true counterfactual $Y_{X'}(T)$. This set may be used to inform better decisions by the user. For instance, an excessively large and diverse set may indicate an unreliable counterfactual estimate. Alternatively, the user may take actions that cater to all plausible counterfactual settings captured by the set (Zecchin et al., 2024).

To achieve this goal, we adapt the **conformal language modeling** (CLM) methodology (Quach et al., 2024) to our counterfactual generation pipeline, thus generalizing CLM to an agentic LLM setting.

## 4.1. Calibration via Test-Time Scaling and Set Prediction

Given a factual episode $T = (X, A, Z, Y)$ and a counterfactual prompt $X'$, the CG method introduced in Sec. 3 is leveraged to produce a sequence of counterfactual estimates $\hat{Y}_{X'}^{(1)}(T), \hat{Y}_{X'}^{(2)}(T), \ldots$ This is done by drawing a sequence of independent samples $\hat{U}_Z^{(k)} \sim q_\phi(U_Z|A, Z)$ over the generation index $k = 1, 2, \ldots$, and using each sample $\hat{U}_Z^{(k)}$ to produce the counterfactual report $\hat{Y}_{X'}^{(k)}(T)$.

At every generation step $k$, write as $C_\lambda^{(k)}$ the current set of $|C_\lambda^{(k)}| \leq k$ generated counterfactual reports, where $\lambda$ are configuration parameters to be introduced in Sec. 4.2. In general, not all previously generated reports are included in set $C_\lambda^{(k)}$. In particular, given $T$, $X'$, and $C_\lambda^{(k-1)}$, the next generated report $\hat{Y}_{X'}^{(k)}(T)$ is included in set $C_\lambda^{(k)}$ according to an acceptance rule $\alpha_\lambda^{(k)} = \alpha_\lambda^{(k)}(T, X', C_\lambda^{(k-1)}, \hat{Y}^{(k)}(T)) \in \{0, 1\}$ so that the set $C_\lambda^{(k)}$ is updated as

$$C_\lambda^{(k)} = \begin{cases} C_\lambda^{(k-1)} & \text{if } \alpha_\lambda^{(k)} = 0 \\ C_\lambda^{(k-1)} \cup \{\hat{Y}_{X'}^{(k)}(T)\} & \text{if } \alpha_\lambda^{(k)} = 1. \end{cases} \quad (4)$$

Furthermore, the procedure stops or continues generating by following a stopping rule $s_\lambda^{(k)} = s_\lambda^{(k)}(T, X', C_\lambda^{(k)}) \in \{0, 1\}$, with $s_\lambda^{(k)} = 1$ indicating that the generation must stop.

The goal of calibration is to select the parameters $\lambda$ such that the generated set $C_\lambda(T, X') = C_\lambda^{(k)}$, with $s_\lambda^{(k)} = 1$, satisfies reliability requirements. To quantify the reliability of set $C_\lambda(T, X')$, let $A : \mathcal{Y} \times \mathcal{Y} \to \{0, 1\}$ be a task-specific admission function in the space $\mathcal{Y}$ of reports, where $A(y_1, y_2)$ equals 1 if $y_1$ is a "good enough" approximation of $y_2$, and 0 otherwise. The admission function $A$ can be obtained by thresholding scores such as the ROUGE distance (Lin, 2004), and embedding-based metrics such as BERTScore (Zhang et al., 2019).

Define the loss for a hyperparameter $\lambda$ on a test point $(T, X')$ as

$$L_\lambda(T, X', Y_{X'}(T)) = \mathbf{1}\{\nexists y \in C_\lambda(T, X') : \\ A(y, Y_{X'}(T)) = 1\}, \quad (5)$$

i.e., the loss $L_\lambda(T, X', Y_{X'}(T))$ equals 1 if there does not exist any report in the generated set $C_\lambda(T, X')$ that is good enough according to the admission function $A$, and 0 otherwise. For a given tolerance $\epsilon > 0$, calibration aims at finding a hyperparameter $\hat{\lambda} \in \Lambda$ ensuring that the set $C_{\hat{\lambda}}(T, X')$ includes at least one report $y \in C_{\hat{\lambda}}(T, X')$ such that $A(y, Y_{X'}(T)) = 1$ with probability at least $1 - \epsilon$, i.e.,

$$R(\hat{\lambda}) = \mathbb{E}\left[L_{\hat{\lambda}}(T, X', Y_{X'}(T))\right] \leq \epsilon. \quad (6)$$

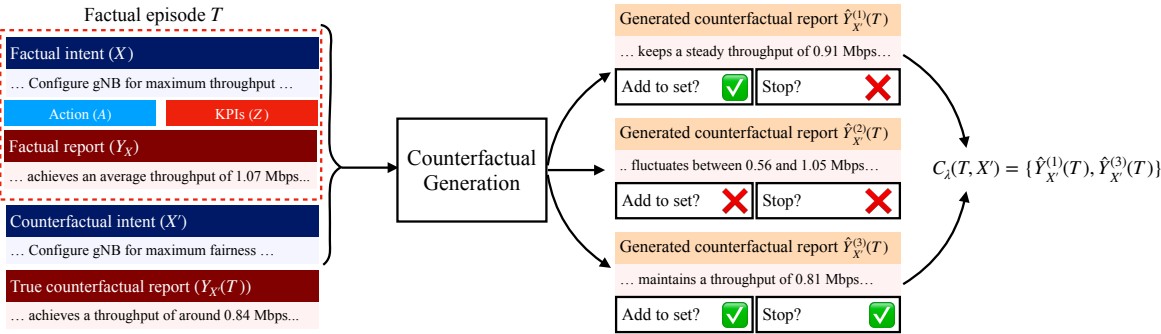

*Figure 4.* Given a factual episode $T = (X, A, Z, Y)$ and a counterfactual intent $X'$, conformal counterfactual generation (CCG) leverages test-time scaling to produce a set $C_\lambda(T, X')$ of counterfactual report candidates with guaranteed reliability with respect to the true counterfactual report.

In (6), the average is evaluated over the joint distribution of episode $T$, counterfactual intent $X'$, and counterfactual report $Y_{X'}(T)$. More precisely, we aim at guaranteeing the inequality

$$\mathbb{P}(R(\hat{\lambda}) \leq \epsilon) \geq 1 - \delta, \tag{7}$$

where $\delta$ is a user-specified outage probability, and the probability is taken over the configuration $\hat{\lambda}$ and thus over the calibration data used to obtain $\hat{\lambda}$ (see Sec. 4.3).

### 4.2. Acceptance and Stopping Rules

In this section, we introduce **conformal CG** (CCG), a calibrated CG strategy that implements acceptance and stopping rules by adapting CLM (Quach et al., 2024).

1) **Acceptance Rule:** The quality function $Q : \mathcal{T} \times \mathcal{X} \times \mathcal{Y} \rightarrow \mathbb{R}$ assigns a numerical score $Q(T, X', \hat{Y}_{X'}^{(k)}(T))$ to each candidate report $\hat{Y}_{X'}^{(k)}(T)$ for a given input $X'$, measuring how well the candidate aligns with the prompt $X'$ and the factual episode $T$. Typical choices include the sequence log-likelihood under the reporting LLM and external verifiers such as LLM-as-a-judge (Gu et al., 2024).

Evaluating the quality of an individual report is not sufficient to judge the utility of a set $C_\lambda(T, X')$, as one also typically wishes the reports in the set to be diverse. To account for this, the similarity function $S : 2^{\mathcal{Y}} \times \mathcal{Y} \rightarrow [0, 1]$ produces a score $S(C_\lambda^{(k-1)}, \hat{Y}_{X'}^{(k)}(T))$ quantifying the similarity of a new candidate report $\hat{Y}_{X'}^{(k)}(T)$ with the reports in set $C_\lambda^{(k-1)}$, with the convention that $S(\varnothing, \cdot) = -\infty$. Common metrics include ROUGE-L$_F$ (Lin, 2004), BLEU (Papineni et al., 2002), or cosine similarity between embedding representations (Mikolov et al., 2013).

Overall, the acceptance function is defined as

$$\begin{aligned} \alpha_\lambda^{(k)} = \mathbf{1}\{ & Q(T, X', \hat{Y}_{X'}^{(k)}(T)) \geq \lambda_1 \text{ and} \\ & S(C_\lambda^{(k-1)}, \hat{Y}_{X'}^{(k)}(T)) \leq \lambda_2 \}, \end{aligned} \tag{8}$$

where hyperparameters $\lambda_1$ and $\lambda_2$ control the quality and

diversity thresholds, respectively.

2) **Stopping rule:** To decide whether to continue generating candidate counterfactual reports, we use a confidence function $F : 2^{\mathcal{Y}} \rightarrow \mathbb{R}$ that aggregates the individual quality scores of the members in the set $C_\lambda^{(k)}$ of candidate reports. While other choices are possible, following (Quach et al., 2024), we choose the max-quality function $F(C_\lambda^{(k)}) = \max_{y \in C_\lambda^{(k)}} Q(T, X', y)$, which focuses on the single best candidate in set $C_\lambda^{(k)}$. Overall, the stopping rule is defined as

$$s_\lambda^{(k)}(T, X', C_\lambda^{(k)}) = \mathbf{1}\{F(C_\lambda^{(k)}) \geq \lambda_3\}, \tag{9}$$

where $\lambda_3$ is the hyperparameter governing the stopping rule.

### 4.3. Conformal Calibration

In order to calibrate the thresholds $\lambda$ that determine the acceptance rule (8) and the stopping rule (9), we leverage a calibration dataset $\mathcal{D}_{\text{cal}}$ of the form $\mathcal{D}_{\text{cal}} = \{(X_i, X_i', T_i, Y_{X_i'}(T_i))\}_{i=1}^n$, where $X_i$ is a factual prompt; each $X_i' \in \mathcal{E}(X_i)$ is a counterfactual prompt obtained by applying an admissible edit to $X_i$; $T_i = (X_i, A_i, Z_i, Y_i)$ is the factual episode; and $Y_{X_i'}(T_i)$ is the true counterfactual report produced by rerunning the environment under $X_i'$ with shared randomness.

Using the calibration dataset $\mathcal{D}_{\text{cal}}$, for a given configuration $\lambda \in \Lambda$, the empirical estimate of the loss (5) is $\hat{R}(\lambda) = \frac{1}{|\mathcal{D}_{\text{cal}}|} \sum_{(T, X') \in \mathcal{D}_{\text{cal}}} L_\lambda(T, X', Y_{X'}(T))$, where $L_\lambda(T, X', Y_{X'}(T))$ is the loss function defined in (5). Define $N_\lambda = |\mathcal{D}_{\text{cal}}| \hat{R}(\lambda)$, so that $N_\lambda$ counts how many calibration pairs are not well covered by the set $C_\lambda(T, X')$ according to the admission function $A$. To decide which configurations $\lambda$ achieve the target miscoverage level $\epsilon$, we map each failure count $N_\lambda$ to a scalar calibration score $p_\lambda \in [0, 1]$ using a binomial tail, following (Angelopoulos et al., 2023; Quach et al., 2024), i.e., $p_\lambda = \sum_{k=0}^{N_\lambda} \binom{n}{k} \epsilon^k (1 - \epsilon)^{n-k}$. Intuitively, if the failure count $N_\lambda$ is smaller than what one

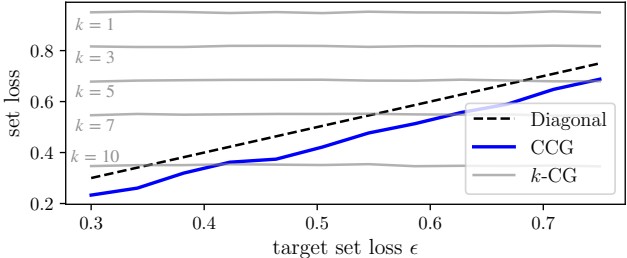

*Figure 5.* Average set loss as a function of the target set loss $\epsilon$ for CCG and fixed-budget baselines $k$-CG. The diagonal indicates ideal risk-controlled behavior.

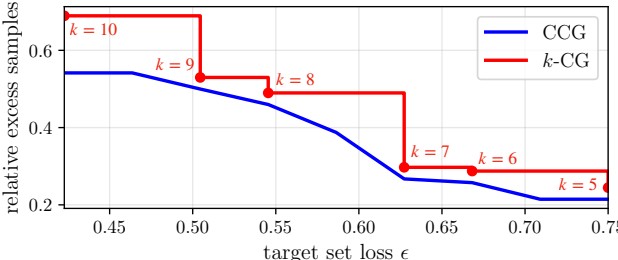

*Figure 6.* Relative excess number of samples as a function of the target set loss $\epsilon$ for CCG and fixed-budget baselines $k$-CG at matched average set sizes.

would typically see under a binomial distribution with parameter $\epsilon$, then the configuration $\lambda$ is deemed sufficiently reliable. Formally, as shown in (Angelopoulos et al., 2023; Quach et al., 2024), the binomial score $p_\lambda$ is a valid p-value for the null hypothesis that hyperparameter $\lambda$ does not satisfy the reliability requirement (7).

Fixing a finite grid $\Lambda$ of candidate configurations, CCG produces the set $\hat{\Lambda} = \{\lambda \in \Lambda : p_\lambda < \bar{p}(\delta)\}$, where the threshold $\bar{p}(\delta)$ is determined by a family-wise error rate (FWER)–controlling procedure at level $\delta$, such as Bonferroni (Bonferroni, 1936) or fixed-sequence testing (FST) (Holm, 1979).

At test time, we ultimately require a single configuration. Following the selection strategy used in CLM (Quach et al., 2024), if set $\hat{\Lambda}$ is empty, we abstain and return no prediction set. Otherwise, any configuration in $\hat{\Lambda}$ is valid, and we select one by minimizing an empirical metric on the calibration data that balances producing compact prediction sets with avoiding unnecessary sampling after a valid candidate has already appeared (see eq. (7) in (Quach et al., 2024)).

Leveraging the same arguments underlying the statistical validity of CLM (Quach et al., 2024), we have the following result (see Appendix F for a proof).

**Proposition 1.** *The set $C_{\hat{\lambda}}(T, X')$ returned by CCG satisfies the reliability guarantee (7).*

### 4.4. Experiments

#### 4.4.1. SETUP AND BENCHMARKS

The experimental setup follows Sec. 3.3, with the caveat that the dataset of 300 data points is split into two disjoint subsets of 150 data points each, one for testing and one for calibration. For the admission function $A$, a candidate counterfactual report $\hat{Y}_{X'}(T)$ is considered admissible if an LLM judge determines that it provides a semantically correct counterfactual for the ground-truth $Y_{X'}(T)$ (see Appendix D for details). Furthermore, following (Quach et al., 2024), the normalized log-likelihood under the report-generating LLM is used as the quality function $Q$. Finally, we adopt the maximum ROUGE–L similarity $S$ between a new candidate

report $\hat{Y}_{X'}^{(k+1)}(T)$ and the current set $C_\lambda^{(k)}$.

We compare the proposed CCG with **fixed-budget baselines** that draw exactly $k$ counterfactual samples without applying any acceptance or stopping logic. These baselines, referred to as $k$-CG, allow us to quantify the efficiency gains from adaptive sampling.

As in (Quach et al., 2024), we report the following three metrics averaged over the dataset and over 50 calibration episodes with different random splits of the dataset: the **set loss** in (5); the **relative excess number of samples** RES $= (k_{\text{stop}} - k^\star)/k^\star$, where $k^\star$ is the index of the first admissible counterfactual report and $k_{\text{stop}}$ is the index at which the sampler terminates; and the **set size** $|C_\lambda(T, X')|$.

#### 4.4.2. RESULTS

We first compare the average set loss achieved by CCG and k-CG with $k \in \{1, 2, \ldots, 10\}$. Fig. 5 reports these quantities as a function of the tolerance level $\epsilon$ in (6). CCG closely follows the diagonal corresponding to risk-controlled behavior, achieving lower set loss as the tolerance level $\epsilon$ increases. In contrast, fixed-budget baselines exhibit either overly conservative or insufficient coverage depending on the sampling budget $k$, with larger values of $k$ reducing the set loss at the cost of increased sampling.

For a given sampling budget $k$, we identify the value of the average set loss $\epsilon$ at which the conformal sampler attains a comparable average set size. This alignment enables a fair comparison of sampling efficiency between the two approaches. As it can be seen in Fig. 6, across all values of $k$, CCG consistently incurs fewer excess samples than the corresponding fixed-budget baselines, with the efficiency gains becoming more pronounced for larger sampling budgets. Additional results illustrating the effect of the calibration set size on set loss and sampling efficiency are reported in Appendix G.

## 5. Conclusion and Future Directions

We have introduced conformal counterfactual generation (CCG), a framework for counterfactual generation in LLM-

powered autonomous control systems, enabling reliable reasoning about alternative user intents in closed-loop agent–environment interactions. Building on conformal language modeling, test-time scaling ensures that CCG meets formal reliability guarantees. Future work can explore extensions to multi-turn and multi-agent settings (Elkael et al., 2025), richer environment models, and interactive human-in-the-loop workflows in which counterfactual analysis is used directly to refine and debug user intents in real time (Amershi et al., 2014).

## Acknowledgments

The work of A. Farzaneh and O. Simeone was supported by the European Research Council (ERC) under the European Union's Horizon Europe Programme (grant agreement No. 101198347). The work of O. Simeone was also supported by an EPSRC Open Fellowship (EP/W024101/1) and by the EPSRC project (EP/X011852/1). The work of S. D'Oro was supported by U.S. NSF under award TI-2449452.

## Impact Statement

This paper presents work whose goal is to advance the field of Machine Learning. There are many potential societal consequences of our work, none which we feel must be specifically highlighted here.

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

## A. Reproducibility

The codebase for the experiments in this paper is publicly available at https://anonymous.4open.science/r/ccg-agentic-counterfactual-9014.

## B. Automatic Generation of Factual and Counterfactual Prompts

To systematically evaluate counterfactual reasoning in the LLM–environment pipeline, we required a diverse set of factual prompts $X$ and their corresponding counterfactual variants $X'$. These prompts were generated automatically using GPT-4 (OpenAI, 2023) following a controlled prompt-editing procedure designed to emulate realistic operator requests in network management.

**Prompt generation procedure:**  Each factual prompt $X$ was produced by conditioning GPT-4 on a high-level description of a network operator's intent, such as configuring traffic conditions, scheduling policy, or duration of a 5G simulation. The model was guided to express the prompt in natural language with realistic phrasing (e.g., "Run a 5G simulation with eight users using the proportional fair scheduler for 10 seconds at 5 Mbps each").

**Counterfactual editing:**  Given each factual prompt $X$, a corresponding counterfactual edit $X'$ was obtained by instructing GPT-4 to modify one or more configuration aspects while keeping the remaining parameters unchanged. Note that this four-parameter restriction is an experimental design choice for controlled evaluation, and does not constitute a limitation of the CCG framework itself, which supports arbitrary edits within $\mathcal{E}(X)$. The controlled edits were restricted to four interpretable parameters:

- **Scheduling policy:** switching between `RR` and `PF`.

- **Number of UEs:** varying within $[3, 10]$.

- **Traffic load:** adjusting the offered per-UE rate between 2 and 10 Mbps.

- **Duration:** changing the simulation length between 5 and 10 seconds.

Edits were generated using a templated system prompt that ensured minimal semantic drift from the original intent:

> "You are generating counterfactual versions of network operator requests. Given a factual request, modify only one or two configuration parameters (e.g., scheduler, number of UEs, traffic, or duration) while keeping the rest identical. The result should remain a realistic and well-phrased 5G simulation instruction."

**Filtering and validation:**  To ensure diversity and internal consistency, the resulting prompt pairs $(X, X')$ were post-processed using rule-based checks that verified syntactic validity and correct parameter extraction. Pairs with ambiguous or conflicting instructions were discarded. The final dataset contains 300 factual–counterfactual prompt pairs, each covering one or more counterfactual dimensions and expressed in natural language suitable for direct use by the LLM agent in the main experiments.

## C. LLM-as-a-Judge Prompt Template

To evaluate the semantic fidelity of the counterfactual reports generated by CG and IG, we adopt an LLM-as-a-judge framework based on (Gu et al., 2024). For each test case, the judge model is provided with the true counterfactual report $Y_{X'}(T)$ and the two candidate reports $\hat{Y}_{X'}^{(CG)}(T)$ and $\hat{Y}_{X'}^{(IG)}(T)$. The model is instructed to read all three reports and select which candidate report (CG or IG) is more semantically aligned with the reference. The prompt template is as follows:

> **System prompt:**
> You are an expert evaluator assessing the quality of AI-generated technical summaries. You will be given one reference report describing a 5G network experiment and two candidate reports generated by different methods. Your task is to decide which candidate report better matches the reference in factual content, structure, and overall semantics.
>
> **User prompt:**
> `Reference Report:` [Insert $Y_{X'}(T)$ here]

```
Candidate A (CG): [Insert Ŷ_{X'}^{(CG)}(T) here]
Candidate B (IG): [Insert Ŷ_{X'}^{(IG)}(T) here]
```

Question: Which candidate report more accurately matches the reference in terms of factual content and descriptive alignment? Answer with only one word: "A" or "B".

The judge's decision (A or B) is parsed automatically to count how many times each method is preferred. To validate the judge, we manually inspected 30 true–estimated counterfactual report pairs and found full agreement between automated judgments and human assessments. We also verified robustness to the choice of admission criterion by re-running the wireless network experiment using bidirectional entailment (Kuhn et al., 2023) as an alternative admission function in place of LLM-as-a-judge. In this alternative, a candidate report $\hat{Y}_{X'}(T)$ is admitted if and only if it mutually entails the reference $Y_{X'}(T)$ in both directions — i.e., $\hat{Y}_{X'}(T) \Rightarrow Y_{X'}(T)$ and $Y_{X'}(T) \Rightarrow \hat{Y}_{X'}(T)$ — as assessed by an NLI model conditioned on the prompt context, with both entailment scores required to exceed a fixed threshold. This bidirectional requirement ensures that admitted candidates faithfully capture the same key facts (scheduler choice, throughput levels, latency behavior, and qualitative trends) as the reference, without being overly permissive. Across all 30 inspected pairs, both admission criteria agreed completely, indicating that the conclusions of the wireless network experiment are not an artifact of the specific judge used.

## D. LLM-as-a-Judge as Admission Function

In the conformal counterfactual experiments of Sec. 4.4, the admission function $A$ introduced in Sec. 4.1 is instantiated using an LLM-as-a-judge mechanism. This appendix describes the prompt template used to implement this admission decision in practice.

For each calibration or test instance, the judge model is provided with the true counterfactual report $Y_{X'}(T)$, together with a single candidate counterfactual report $\hat{Y}_{X'}(T)$ generated by the counterfactual generation pipeline. The judge is asked to assess whether the candidate report constitutes a semantically correct description of the reference counterfactual outcome.

The prompt template used for this purpose is as follows.

> **System prompt:**
> You are an expert evaluator assessing the quality of AI-generated technical counterfactual reports for 5G network experiments. You will be given one reference report describing the true counterfactual outcome of an experiment, and one candidate report generated by a counterfactual generation mechanism. Your task is to determine whether the candidate report provides a semantically correct and faithful description of the reference outcome.
>
> **User prompt:**
> ```
> Reference Counterfactual Report: [Insert Y_{X'}(T) here]
> Candidate Counterfactual Report: [Insert Ŷ_{X'}(T) here]
> ```
>
> Question: Does the candidate report accurately capture the key factual and semantic aspects of the reference report (e.g., scheduler choice, throughput levels, latency behavior, and qualitative trends)? Answer with only one word: YES or NO.

This LLM-as-a-judge framework is used as the admission function $A$ for the experiment of Sec. 4.4.

## E. Details of the Abduction Step

Specifically, we construct a training dataset by first drawing $U_Z \sim p(U_Z)$ from the prior and actions $A \sim p(A)$ from a dataset of factual actions. Then, we compute the corresponding observations $Z = f_Z(A, U_Z)$ using the model (3). The resulting triplets $(A, Z, U_Z)$ are then used to train the model $q_\phi(U_Z \mid A, Z)$ by maximum likelihood, i.e., by maximizing the log-likelihood of the latent variables $U_Z$ under the learned conditional density (Greenberg et al., 2019; Papamakarios et al., 2019).

Once trained, $q_\phi(U_Z \mid A, Z)$ enables efficient amortized inference of the exogenous noise variable $U_Z$ for new factual pairs $(A, Z)$. In particular, the outcome of abduction is a noise realization $\hat{U}_Z \sim q_\phi(U_Z \mid A, Z)$ sampled from the estimated posterior (Cranmer et al., 2020; Papamakarios et al., 2019).

To train $q_\phi(U_Z \mid A, Z)$, we generated an auxiliary dataset of triplets $(A, Z, U_Z)$ by randomly sampling gNB configurations $A$ from a uniform prior over the number of UEs (3–10), schedulers (RR/PF), per-UE traffic load (2–10 Mbps), and observation

duration (5–10 seconds), and recording the simulator output $Z = f_Z(A, U_Z)$ under independently sampled noise seeds $U_Z$. The model $q_\phi(U_Z \mid A, Z)$ is implemented as a fully connected neural network with three hidden layers of 128 units and ReLU activations, and is trained using the Adam optimizer by minimizing the negative log-likelihood loss

$$\mathcal{L}(\phi) = -\frac{1}{N} \sum_{j=1}^{N} \log q_\phi(U_{Z,j} \mid A_j, Z_j), \tag{10}$$

where $\{(A_j, Z_j, U_{Z,j})\}_{j=1}^{N}$ denotes the generated training dataset (Greenberg et al., 2019; Papamakarios et al., 2019).

## F. Proof of Proposition 1

*Proof.* The binomial scores $p_\lambda$ constructed in Sec. 4 are valid calibration scores for the miscoverage events defined in (6), as shown in (Angelopoulos et al., 2023; Quach et al., 2024). A level-$\delta$ FWER-controlling procedure applied to $\{p_\lambda : \lambda \in \Lambda\}$ ensures that, with probability at least $1 - \delta$, every configuration in $\hat{\Lambda}$ attains miscoverage probability at most $\epsilon$. On this event, the guarantee (7) holds simultaneously for all $\lambda \in \hat{\Lambda}$, and therefore also for the selected configuration $\hat{\lambda}$. $\qquad\square$

## G. Effect of the Calibration Set Size

In this appendix, we provide additional experimental results that complement Sec. 4.4 by studying the effect of the size of the calibration dataset on the behavior of CCG.

Specifically, we fix the target set loss to $\epsilon = 0.5$ and vary the number of calibration samples $n_{\text{cal}}$ from 5 to 100, while keeping the remaining experimental setup identical to that described in Sec. 4.4. For each value of $n_{\text{cal}}$, we evaluate the average set loss and the relative excess number of samples, averaged over 20 random splits of the dataset.

**Set loss:** Fig. 7(a) reports the average set loss as a function of the calibration set size. As expected from CCG, increasing the number of calibration samples does not substantially reduce the achieved set loss beyond the target level $\epsilon$. Instead, the set loss remains tightly concentrated slightly below $\epsilon$ across all values of $n_{\text{cal}}$, confirming that calibration primarily serves to enforce reliability rather than to optimize predictive accuracy.

**Sampling efficiency:** In contrast, Fig. 7(b) shows a clear and systematic improvement in sampling efficiency as the calibration set size increases. Specifically, the relative excess number of samples decreases markedly with $n_{\text{cal}}$, reflecting more accurate calibration of the acceptance and stopping rules. Nevertheless, the relative excess does not vanish even for large calibration sets. This behavior captures the intrinsic variability of the test-time sampling process and the fact that, even with a larger calibration dataset, additional samples may be required before a high-quality counterfactual report is identified.

Overall, these results highlight that the primary benefit of increasing the calibration set size in CCG lies in improved sampling efficiency rather than in further reductions of the set loss, which remains controlled near the prescribed tolerance level $\epsilon$.

## H. Effect of Simulator Quality

A key practical consideration in simulation-in-the-loop counterfactual reasoning is the quality (or fidelity) of the simulator used in the environment model $Z = f_Z(A, U_Z)$. In our setup, the real environment is emulated in ns-3 with a packet-level 5G NR stack, including fast fading, shadowing, and stochastic traffic/queueing dynamics, while the digital-twin simulator $f_Z$ intentionally adopts simplified abstractions (Sec. 3.3). This appendix studies how the channel-model realism of the simulator impacts (i) the CG fidelity at the KPI level, and (ii) the sampling efficiency of CCG.

### H.1. Simulator-quality via discrete channel-model fidelity levels

To study the effect of simulator fidelity in a precise and reproducible manner, we introduce a discrete simulator quality index $Q \in \{1, 2, 3, 4\}$, where larger values correspond to higher-fidelity channel modeling in the digital-twin simulator $f_Z$. Each quality level $Q$ is associated with a specific ns-3 channel-model configuration, and transitions between quality levels correspond to enabling additional sources of physical-layer realism.

The four simulator quality levels considered in this section are defined as follows.

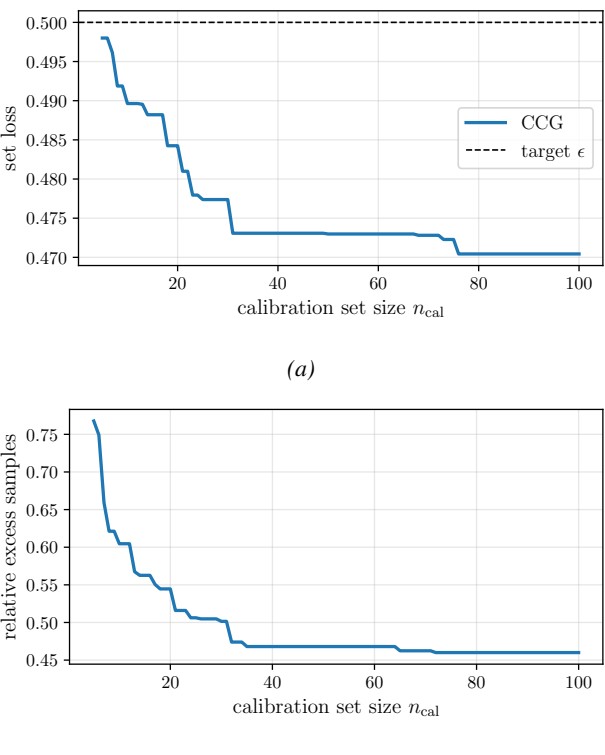

*(a)*

*(b)*

*Figure 7.* Effect of calibration set size on CCG for fixed target set loss $\epsilon = 0.5$ (averaged over 20 random dataset splits). **(a)** Average set loss as a function of $n_{\text{cal}}$, which remains tightly concentrated slightly below $\epsilon$ across calibration sizes. **(b)** Relative excess number of samples as a function of $n_{\text{cal}}$, which decreases substantially with calibration size but converges to a nonzero level due to inherent variability in the stopping time.

- **Level $Q = 1$ (Low fidelity):** The simulator uses a deterministic averaged link-gain channel model based on distance-dependent path loss only. Fast fading and shadowing are disabled, and the scheduler observes a temporally smooth SINR process. This level corresponds to a highly abstract channel model that suppresses short-term variability.

- **Level $Q = 2$ (Medium–low fidelity):** Log-normal shadowing is added on top of path loss, introducing slow spatial and temporal variability in the channel. Fast fading remains disabled. Compared to $Q = 1$, this level captures large-scale channel uncertainty but still averages out small-scale effects.

- **Level $Q = 3$ (Medium–high fidelity):** Independent fast fading (Rayleigh) is enabled in addition to path loss and shadowing. This introduces realistic short-term channel fluctuations that directly affect scheduling decisions and per-UE throughput dynamics.

- **Level $Q = 4$ (High fidelity):** The simulator employs a full stochastic channel model including path loss, shadowing, and fast fading, with fine-grained temporal updates at the scheduler. This implements the same dynamics as the real environment.

Across all quality levels $Q$, all other components of the simulator (including scheduler logic, traffic model, and observation granularity) are kept fixed. This ensures that performance variations observed in the following experiments can be attributed primarily to changes in channel-model realism.

### H.2. Impact on counterfactual generation fidelity

Fig. 8a reports the reconstruction error of the counterfactual KPI time series as a function of the simulator quality level $Q$. For both per-UE throughput and end-to-end delay, the MAE decreases monotonically as the simulator fidelity increases, with the most pronounced improvements observed when moving from low to intermediate quality levels. The gains then gradually saturate as the simulator approaches the highest-fidelity configuration.

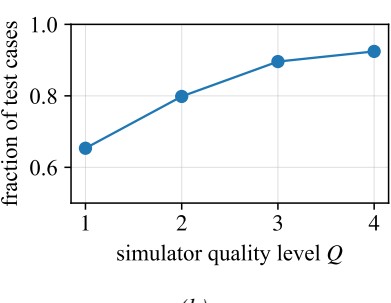
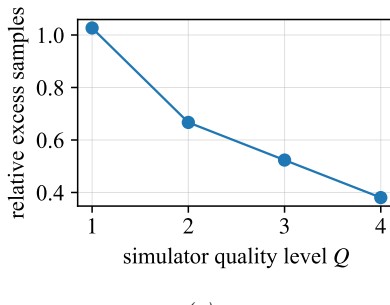

*Figure 8.* Effect of simulator quality on counterfactual generation and conformal calibration. The x-axis in all panels reports the discrete simulator quality level $Q$, from low to high channel-model fidelity. **(a)** KPI-level counterfactual fidelity, measured by the mean absolute error (MAE) of the reconstructed counterfactual KPI time series for throughput and delay. **(b)** Semantic fidelity of counterfactual reports, measured as the fraction of test cases in which CG is selected as the best semantic match to the true counterfactual report by an LLM judge. **(c)** Conformal sampling efficiency, measured by the relative excess number of samples (RES) at a fixed target set loss $\epsilon = 0.5$.

This trend can be explained by the role of the simulator in the abduction step of counterfactual generation. CG infers the latent environment noise $U_Z$ by learning an approximate posterior $q_\phi(U_Z \mid A, Z)$ from samples generated by the simulator. At low fidelity, the simulator $f_Z(A, U_Z)$ fails to capture key stochastic properties of the real environment trace $Z$, leading to a model mismatch that biases the inferred noise realization $\hat{U}_Z$. As the simulator quality increases, this mismatch is progressively reduced, yielding more accurate abduction and, consequently, more faithful reconstructions of the counterfactual KPI sequence $\hat{Z}_{X'}(T) = f_Z(A_{X'}(T), \hat{U}_Z)$. In this regard, it is noted that the performance at the highest-quality level $Q = 4$ reflects the degradation caused by an independent estimate of the exogenous variable $\hat{U}_Z$.

In addition to KPI-level fidelity, we evaluate how simulator quality affects the semantic quality of the counterfactual reports generated by CG. For each simulator quality level $Q$, we compare the estimated counterfactual report $\hat{Y}_{X'}(T)$ produced by CG against the true counterfactual report $Y_{X'}(T)$ using the same LLM-as-a-judge protocol adopted in the main text (Sec. 3.3). The judge is provided with the true counterfactual report as a reference and is asked to select which candidate report best matches it in content and semantics.

Fig. 8b reports the fraction of test cases in which the counterfactual report generated by CG is selected as the best semantic match to the true counterfactual report, as a function of the simulator quality level $Q$. The results indicate a clear improvement in semantic fidelity as simulator quality increases. At low fidelity levels, CG-generated reports are more frequently judged to deviate from the true counterfactual outcome, reflecting distortions in the underlying counterfactual environment rollouts. As simulator realism improves, the fraction of cases in which CG is preferred increases steadily, approaching the performance observed in the main experimental setting.

### H.3. Impact on conformal sampling efficiency

Fig. 8c shows the relative excess number of samples required by CCG as a function of the simulator quality level $Q$, for a fixed target set loss $\epsilon = 0.5$. As the simulator fidelity increases, the RES consistently decreases, indicating that the conformal sampler terminates earlier and requires fewer additional samples beyond the first admissible counterfactual report.

This behavior follows from how simulator fidelity affects the distribution of counterfactual candidates generated during test-time scaling. When the simulator is low fidelity, the induced counterfactual feedback $\hat{Z}_{X'}^{(k)}(T)$ may exhibit systematic distortions, such as overly smoothed channel dynamics or missing variability. These distortions propagate to the report generator and reduce the probability that an individual candidate satisfies the task-specific admission function $A$ (Appendix D), thereby delaying the stopping condition. Improving simulator fidelity mitigates these effects, increasing the likelihood that high-quality counterfactual reports are generated earlier and enabling more efficient conformal stopping.

## I. Drone Navigation Experiment

To assess the generality of CG and CCG beyond the wireless network control setting, we present an additional experiment in a drone navigation domain, using a different environment, a different LLM, and a broader edit space that explicitly separates intent changes from paraphrases.

### I.1. Setup

**Environment:** We consider a drone navigation task on a $16 \times 16$ 2D grid. The drone starts from a known initial location and must reach a target cell under open-loop control. The map contains three region types: (i) *safe cells*, where motion is deterministic; (ii) *windy cells*, where the drone experiences state-dependent stochastic perturbations governed by a wind field; and (iii) *keep-out cells*, which the drone must avoid. The wind field constitutes the exogenous noise $U_Z$, with dynamics $Z = f_Z(A, U_Z)$ as in the main framework. An example environment is illustrated in Fig. 9.

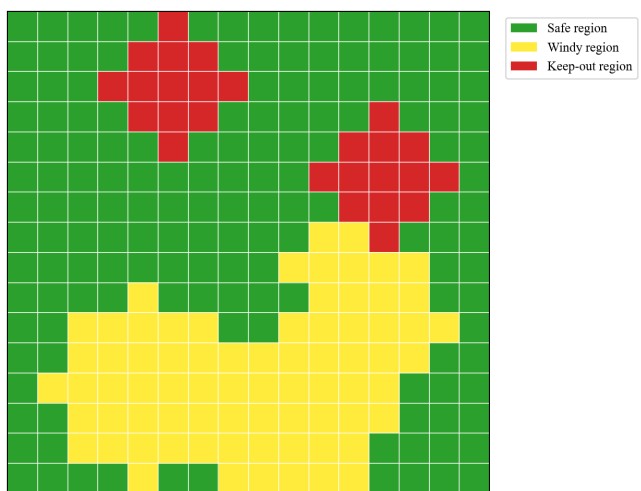

*Figure 9.* Example $16 \times 16$ drone navigation environment. Safe cells (white), windy cells (blue), and keep-out cells (red) are shown. The drone starts at the green cell and must reach the target (yellow cell) while avoiding keep-out zones.

**Counterfactual edit space:** We construct counterfactual prompts of three types to explicitly separate different intent from different wording:

- **Preference change**: e.g., replacing "prioritizing safety" with "prioritizing speed."

- **Target/constraint change**: e.g., changing the goal cell from $(7, 8)$ to $(5, 5)$.

- **Paraphrase**: expressing the same intent and parameters with different wording (e.g., "Fly to $(7, 8)$ in 20 moves" vs. "Navigate the drone to target $(7, 8)$ within 20 steps").

**LLM agent:** We use Qwen2 7B Instruct (Yang et al., 2024) as the LLM agent, demonstrating that the method is not tied to any specific model.

**Admission function:** We replace the LLM-as-a-judge admission criterion with a semantics-aware rule based on bidirectional entailment, following the semantic entropy framework of (Kuhn et al., 2023). Specifically, two reports are considered semantically equivalent if they mutually entail each other: a candidate report $\hat{Y}_{X'}(T)$ is admitted if and only if both $\hat{Y}_{X'}(T) \Rightarrow Y_{X'}(T)$ and $Y_{X'}(T) \Rightarrow \hat{Y}_{X'}(T)$ hold, as assessed by a natural language inference (NLI) model conditioned on the shared prompt context (i.e., the initial drone state, goal cell, and environment layout). Both entailment directions are required: one-way entailment is insufficient, as a report that describes a superset of the reference content would pass a one-directional check without faithfully capturing the counterfactual outcome. Admission is granted only when both entailment scores exceed a fixed threshold, ensuring that the candidate and reference reports describe the same trajectory outcome, constraint violations, and success/failure status, regardless of surface-level phrasing differences.

### I.2. Results: Counterfactual Generation Fidelity

We evaluate trajectory-level accuracy via the mean absolute error

$$\text{MAE}_{\text{traj}} = \frac{1}{H} \sum_{t=1}^{H} \left\| s_{X'}^{(t)}(T) - \hat{s}_{X'}^{(t)}(T) \right\|_1, \tag{11}$$

which measures the average per-step deviation between the predicted and true counterfactual trajectory positions over a horizon of $H$ steps. Table 2 reports results for IG, SIG, and CG across the three edit types. CG consistently achieves the lowest trajectory error, with the largest gains for preference and target/constraint changes. For paraphrases, all methods produce similar (low) errors, as expected since the intent is preserved.

*Table 2.* Trajectory MAE ($\downarrow$) in the drone navigation experiment (best in **bold**).

| Edit type | CG | IG | SIG |
|---|---|---|---|
| Preference change | **1.74** | 3.84 | 2.96 |
| Target/constraint change | **2.08** | 4.27 | 3.41 |
| Paraphrase | **0.36** | 1.21 | 0.88 |

## I.3. Results: Simulator Fidelity Ablation

We introduce a simulator fidelity axis by varying the spatial resolution of the digital twin's transition model. As shown in Fig. 10, both trajectory MAE and admission rate improve monotonically with simulator resolution, consistent with the findings of Appendix H.

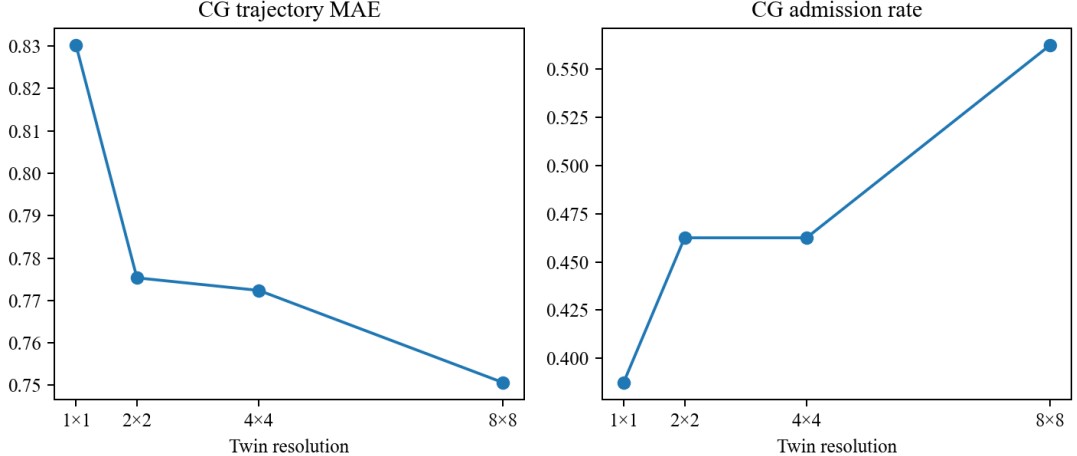

*Figure 10.* Effect of simulator resolution on CG trajectory fidelity (MAE, left axis) and CCG admission rate (right axis) in the drone navigation experiment.

## I.4. Computational Cost and Scaling

The total computational cost of CCG has three components. The **offline** cost covers training the NPE posterior $q_\phi(U_Z \mid A, Z)$ and running conformal calibration on $\mathcal{D}_{\text{cal}}$, both performed once before deployment and scaling linearly with dataset size. The **per-query** cost at test time requires $k_{\text{stop}}$ simulation rollouts, each of cost $C_{\text{sim}}$, giving a total of $O(k_{\text{stop}} \cdot C_{\text{sim}})$; CCG reduces $k_{\text{stop}}$ relative to fixed-budget baselines by terminating as soon as a high-quality candidate is found. Finally, tighter reliability requirements (smaller $\epsilon$ or $\delta$) may increase $k_{\text{stop}}$; if real-time response is needed and $C_{\text{sim}}$ is large, a learned surrogate can replace the full simulator at the cost of reduced abduction fidelity (cf. Appendix H). In the wireless network experiment, each ns-3 simulation takes approximately 2–5 seconds on a standard CPU, and the CCG sampler requires on average fewer than 4 rollouts to terminate at $\epsilon = 0.5$, making it practical for offline analysis.

