# OpenReview forum: "Should I Have Expressed a Different Intent? Counterfactual Generation for LLM-Based Autonomous Control"
_ICML.cc/2026/Conference — ICML 2026 regular_

### Official Review · Reviewer_8Nkh · 2026-03-05

**Soundness:** 3
**Presentation:** 3
**Significance:** 3
**Originality:** 3
**Overall Recommendation:** 5
**Confidence:** 3

**Summary:**

The paper addresses the problem of reliable counterfactual generation in LLM-based systems, specifically attempting to answer how to edit the prompt in a way that captures the different design goals and environments of the user within the system. The method first creates a set of actions using structured outputs, then a report generator is used to provide an initial report. After the report is generated, the counterfactual generation produces new prompts by recovering the “latent randomness” of the initial report via neural posterior estimation and generates counterfactual actions. The authors benchmark their results on a 5G network setup and provide reliability guarantees on the generated counterfactual prompts.

**Compliance With Llm Reviewing Policy:**

Affirmed.

**Final Justification:**

The rebuttal did address and my and the other reviewers' concerns. I think the authors did a good job in addressing the concerns in general, and the paper is solid for its application.

**Key Questions For Authors:**

1- How sensitive are the calibration results to the choice of judge? If a different judge model or a different prompt template were used, would the thresholds change substantially?

2- As I mentioned in my comments, the set of candidate grid configurations is finite. How sensitive are the results with the grid selection, and does the method scale to finer or continuous grid resolutions?

3- The paper uses an old LLM (Llama 3 8b-Instruct) as the LLM agent, do the results change with more recent LLMs with similar scale?

4- The structured causal model assumes that the exogenous variables for the action (A), report (Z), and the feedback (Y) are independent of the prompt (X). I understand that this assumption is needed for the reliability proofs, but is this a valid assumption for the practical settings where the user’s prompt can be correlated with the report and feedback?

5- The counterfactual prompts only consider edits of four parameters. Is it possible to relax this assumption in a practical way?

**Limitations:**

yes

**Strengths And Weaknesses:**

I like the approach of providing a causal inference framework in trying to estimate the environment noise to guarantee that the resulting set of prompts approximate the actual counterfactual report with high propability.

I also think that the approach of providing reliability guarantees instead of just estimating success rate is a meaningful contribution.

Finally, the authors show strong results on the 5G setting compared to the other two baselines with running an independent full episode with the update prompt and using the simulator instead of the environment. I also appreciate the ablation results with the simulator quality ablation.

I think the main weakness of the paper is having a single network task with a set of discrete action choices, which makes the calibration of the counterfactual prompts much more tractable. I do acknowledge that it may be impossible to get formal guarantees with continuous action spaces, but I think a numerical example would be useful to obtain, even though there are no formal guarantees.

As the authors mention in their ablations, the method relies on a simulator that is well calibrated, and the authors should benchmark their results with setups that may have bigger gaps between the simulator and reality, for example, with partial observation of the action outcomes, or having noisy estimates of the results.

Finally, the LLM-judge setup is useful to automate the method, but it may be hard to verify the results of the judge.

---

> ### Author Rebuttal · Authors · 2026-03-27
>
> We thank the reviewer for the thoughtful and constructive review. We address the questions below.
>
> **1) Sensitivity of calibration to the judge / admission rule**
> Please see points 1 and 2 of our response to Reviewer WSHx.
>
> **2) Sensitivity to the finite candidate grid and continuous resolutions**
> Yes, the current implementation calibrates over a finite grid $\Lambda$, following the conformal language modeling framework we build upon [1]. A finer grid improves resolution but increases computational cost and the multiplicity burden in model selection. Thus, there is a standard trade-off between expressivity and statistical efficiency. Extending the method to continuous parameterizations is an interesting direction, but the finite-grid setup allows us to maintain transparent and tractable guarantees. We will clarify this limitation in the revision.
>
> **3) Use of Llama 3 8B-Instruct model**
> We chose Llama 3 8B-Instruct as a strong open model that enables repeated sampling, ablation studies, and conformal calibration at a manageable computational cost. Importantly, our method is model-agnostic: the SCM, abduction, replay, and conformal wrapper do not depend on the specific base model. To further support this, we have included a new experiment using Qwen2 7B Instruct (see response to Reviewer pzr9), demonstrating consistent behavior across different LLMs of similar scale.
>
> **4) Independence of exogenous variables from the prompt**
> We clarify that this is not an ad hoc assumption, but standard SCM semantics. By the functional representation lemma [5], any conditional distribution of $Y$ given $X$ can be represented as $Y = f(X,U)$ with $U$ independent of $X$; this underlies any SCM [3–4]. In our setting, dependence between the prompt $X$ and downstream variables is captured by the structural functions $f_A, f_Z, f_Y$, while the exogenous variables represent residual randomness and can be taken independent by construction; see [3–4]. We will clarify this point in the revision.
>
> **5) Restricting edits to four parameters**
> The framework itself does not require this restriction. The four-parameter edit space was a controlled experimental design choice to enable systematic factual/counterfactual comparisons. In the new robotics experiment (see response to Reviewer pzr9), we consider a broader and more flexible edit space, demonstrating that the method extends beyond this setting.
>
> **6) Broader empirical scope beyond a single discrete task**
> Please see our response to Reviewer pzr9, where we introduce a second experiment in a different domain.
>
> We thank the reviewer again for the constructive feedback.
>
> ---
>
> **References**
>
> [1] Quach, V. et al., “Conformal Language Modeling”, 2024.
>
> [2] Kuhn, L., Gal, Y., and Farquhar, S., “Semantic Uncertainty: Linguistic Invariances for Uncertainty Estimation in Natural Language Generation”, 2023.
>
> [3] Pearl, J., “Causality: Models, Reasoning, and Inference”, 2009.
>
> [4] Peters, J., Janzing, D., and Schölkopf, B., “Elements of Causal Inference”, 2017.
>
> [5] Kallenberg, O., “Foundations of Modern Probability”, 1997.

---

> > ### Author Rebuttal · Reviewer_8Nkh · 2026-04-02
> >
> > Thanks a lot for the detailed response, I am happy to raise my score to accept.

---

### Official Review · Reviewer_pzr9 · 2026-03-06

**Soundness:** 3
**Presentation:** 3
**Significance:** 2
**Originality:** 3
**Overall Recommendation:** 4
**Confidence:** 5

**Summary:**

This paper addresses counterfactual reasoning in LLM-based autonomous control systems, where a user may wish to understand how an alternative phrasing of intent would have changed the resulting behaviour. The authors model the closed-loop interaction between user prompt, LLM agent, environment, and generated report as a structural causal model, enabling counterfactual inference through abduction and replay under modified intents. The framework produces alternative action sequences, environment trajectories, and resulting reports corresponding to edited user prompts.
To manage uncertainty in counterfactual estimation, the paper introduces Conformal Counterfactual Generation (CCG). Rather than returning a single counterfactual outcome, the method produces a calibrated set of candidate outcomes and applies a conformal calibration procedure to provide statistical coverage guarantees. Under this formulation, the generated set is designed to contain a semantically faithful approximation of the true counterfactual outcome with high probability. Experiments in a wireless network control setting demonstrate improved fidelity compared to interventional baselines, while the conformal mechanism maintains reliable coverage with controlled sampling cost.
Overall, the paper presents a coherent integration of causal modelling, simulation-based counterfactual inference, and conformal calibration for analysing alternative intents in agentic LLM systems.

**Compliance With Llm Reviewing Policy:**

Affirmed.

**Final Justification:**

I still see the contribution as somewhat more integrative than foundational, and I think the broader generality claims should continue to be stated with some care. I also remain slightly cautious about the robustness of the semantic evaluation setup in general. Overall, however, the paper is technically coherent, the rebuttal resolves the main issues I raised, and I think the submission clears the bar for acceptance. I therefore maintain my positive score.

**Key Questions For Authors:**

1. Generality beyond the wireless control domain
Your framework is presented as a general method for counterfactual reasoning in agentic LLM systems, but the evaluation is confined to a single simulated wireless control environment. How well do you expect the approach to transfer to substantially different settings (e.g., non-simulator domains, open-ended planning tasks, or partially observable environments)? Are there particular domain properties—such as simulator fidelity, controllability, or stochasticity—under which the method is expected to succeed or fail? A clearer discussion or preliminary evidence here would strengthen the claim of generality. I want to believe, but I don’t see enough evidence.

2. Sensitivity to simulator mismatch
The method depends on a digital twin of the environment to generate counterfactual trajectories. Could you elaborate on how sensitive the approach is to simulator inaccuracies? After all, by definition, simulators are wrong unless there is something very unusual about it. For example, if the simulator is mildly misspecified, how does this affect the counterfactual fidelity and validity of the conformal coverage guarantee in practice? THis should be addressed

3. Nature of the conformal guarantee
The conformal procedure guarantees coverage relative to the admission function used to accept counterfactual candidates. Could you clarify how sensitive the guarantee is to the choice of this function (e.g., semantic similarity metric or LLM judge)? In particular, how should one interpret the guarantee when the admission criterion is only an approximate proxy for true counterfactual correctness?

4. Evaluation of semantic fidelity
Semantic correctness is evaluated primarily using automated metrics and LLM-based judging. Did you consider human or task-grounded validation of counterfactual quality, even on a small subset? If so, what were the outcomes? If not, how confident are you that the automated evaluation accurately reflects meaningful counterfactual fidelity? This is real weak point in the paper.

5. Computational cost and scaling behaviour
The framework relies on repeated abduction and sampling, together with conformal calibration. Could you provide more insight into how computational cost scales with environment complexity, simulator cost, and required confidence level? In particular, is the approach practical in settings where simulation is expensive or real-time response is required?

**Limitations:**

The authors do discuss several important limitations, including the reliance on a simulator (digital twin) of the environment, the conditional nature of the conformal coverage guarantee with respect to the chosen admission function, and the computational trade-offs introduced by repeated sampling and calibration. These points are helpful and demonstrate appropriate awareness of the method’s practical constraints.

That said, some aspects could be discussed more fully. In particular, the sensitivity of the approach to simulator misspecification, the extent to which the framework generalises beyond the evaluated domain, and the reliance on automated/LLM-based assessment of semantic fidelity would benefit from deeper analysis. A clearer discussion of these issues would strengthen the paper and help clarify the conditions under which the method is expected to perform reliably.

The paper doesn’t raise any significant  societal concerns, although broader deployment in decision-making or control settings would require careful consideration of robustness, interpretability, and reliability under model mismatch, especially as there’s no human in the loop at the moment.

**Strengths And Weaknesses:**

This is a thoughtful and technically coherent paper addressing an increasingly relevant question: how to reason about alternative user intents in LLM-driven autonomous systems. The authors present a principled framework grounded in structural causal modelling and augment it with conformal calibration to provide statistical reliability. The work is carefully constructed and clearly presented, and it reflects a solid understanding of both causal inference and uncertainty quantification. Overall, there is real substance here.

What I liked

Conceptual clarity and grounding.

The framing of the agent–environment–user interaction as a structural causal model is natural and well motivated. The abduction-and-replay formulation for counterfactual reasoning is technically sound and provides a coherent basis for your proposed method.

Integration of causal inference with calibrated uncertainty.
The incorporation of conformal prediction — this was new too me and I like it —  is a strong aspect of the work. Rather than producing a single speculative counterfactual, the method generates calibrated sets with coverage guarantees, which is particularly valuable in decision-making and control settings. This adds meaningful statistical grounding beyond heuristic counterfactual generation.

Technical care and coherence.
The paper is methodologically careful. The modelling assumptions are clearly stated, the probabilistic formulation is consistent, and the conformal calibration procedure is appropriately constructed. The system hangs together well, both conceptually and technically.

Clear and readable presentation.
The paper is well written and logically organised. It was an easy and natural lead, although I was slightly concerned at how overly informal it souned at the start, but that wasn’t typical of the rest, in fairness. The narrative progresses cleanly from motivation to modelling to method and evaluation. The exposition of the conformal component, in particular, is clear and accessible despite the technical content.

Demonstrated improvement over simple baselines.
The experimental results show that the approach produces more faithful counterfactual outcomes than naive interventional or re-execution strategies. The reliability–efficiency trade-off of the conformal layer is also illustrated effectively.


What I liked less

Narrow empirical scope.
This is the biggest issue: The evaluation is conducted entirely within a single application domain (wireless network control). While this is a meaningful and non-trivial setting, it leaves open the question of how broadly the framework applies. Since the method is presented as a general approach to counterfactual reasoning in agentic LLM systems, evidence from additional domains or task types would strengthen the claim of generality.It’s a shame, becuyase I really wanted to like the paper!

Dependence on simulation fidelity.
The method relies on a digital twin of the environment to generate and evaluate counterfactual trajectories. As with any simulation-based counterfactual approach, the quality of the results depends on how accurately the simulator reflects reality. This limitation is acknowledged but remains an important practical consideration.

Evaluation of semantic fidelity.
Semantic correctness of counterfactual reports is assessed using automated metrics and LLM-based judging. While this is increasingly common, it introduces some uncertainty regarding the true semantic quality of generated counterfactuals. Complementary human or task-grounded validation would have strengthened confidence in these results.

Moderate novelty at the component level.
Many individual ingredients of the approach—SCM-based counterfactual reasoning, likelihood-free abduction, and conformal calibration—are established. The contribution lies primarily in their integration and application to agentic LLM control. This is a meaningful contribution, but it is more integrative than foundational.

Limited discussion of broader applicability.
The paper would benefit from a much deeper discussion of where the framework is expected to generalise well, where it may struggle, and what properties of a domain (e.g., simulator quality, controllability, stochasticity) are required for reliable operation.

---

> ### Author Rebuttal · Authors · 2026-03-27
>
> We sincerely thank the reviewer for the detailed, thoughtful, and highly constructive review. Below we address the main concerns.
>
> **1) Generality beyond wireless control**
> To directly address the reviewer’s main concern, we implemented an additional experiment in a different domain inspired by recent work on reliable LLM-guided human-autonomy teaming [2].
>
> We consider a drone-navigation task on a 2D grid map. The drone starts from a known initial location and must reach a target under open-loop control. The map contains three region types: (i) safe cells (deterministic motion), (ii) windy cells (state-dependent stochastic perturbations), and (iii) keep-out cells. The wind field acts as exogenous noise $U_Z$, with dynamics
> $$
> s_{t+1} = s_t + a_t + \epsilon(s_t).
> $$
> An example environment is available here: https://anonymous.4open.science/r/Additional-Results-6249/fig_map.png
>
> An example of a factual prompt is:
> “Navigate the drone to target (7,8) within 20 steps while prioritizing safety and avoiding restricted zones.”
>
> We construct counterfactual prompts of three types:
> - Preference change: “...prioritizing speed instead of safety.”
> - Target/constraint change: “...to target (5,5)...”
> - Paraphrase: “Fly to (7,8) in 20 moves, avoid restricted zones, and prioritize safety.”
>
> This explicitly separates different intent from different wording (point 3 of response to Reviewer WSHx).
>
> We use Qwen2 7B Instruct (different from the wireless experiment), showing the method is not tied to a specific LLM.
>
> We evaluate trajectory accuracy via
> $$
> \mathrm{MAE} = \frac{1}{T} \sum_{t=1}^T \| s_t^{\mathrm{cf}} - \hat{s}_t \|_1,
> $$
> which measures the average step-wise deviation between the predicted and true counterfactual trajectories, i.e., how far the simulated path drifts from the true rollout over time.
>
> **Trajectory error results (MAE; lower is better):**
>
> | Method | Preference change | Target/constraint change | Paraphrase |
> |--------|------------------|--------------------------|------------|
> | IG     | 3.84             | 4.27                     | 1.21       |
> | SIG    | 2.96             | 3.41                     | 0.88       |
> | CG     | **1.74**         | **2.08**                 | **0.36**   |
>
> CG consistently achieves the lowest error, with largest gains when plans change.
>
> We further introduce a simulator-fidelity knob by varying the spatial resolution of the twin’s transition model. Specifically, we construct twins at progressively finer resolutions, where higher resolution corresponds to a model that more closely matches the true 16x16 environment dynamics. We report both trajectory MAE and admission rate (the fraction of semantically admissible reports). Results are provided at: https://anonymous.4open.science/r/Additional-Results-6249/fig_quality_ablation.png.
> As the resolution increases, both geometric accuracy and semantic validity improve monotonically, confirming that the method benefits from higher-fidelity twins.
>
> Importantly, in this experiment we also replace the LLM admission function with a semantics-aware alternative (see point 4), demonstrating that the framework is flexible and not tied to a specific evaluation mechanism.
>
> **2) Sensitivity to simulator mismatch**
> This is directly studied via the fidelity ablations in both experiments. Reduced fidelity degrades accuracy (MAE, semantic agreement), but the conformal guarantee remains valid relative to the admission function.
>
> **3) Nature of the conformal guarantee**
> Please refer to our response to Reviewer WSHx (Point 1).
>
> **4) Evaluation of semantic fidelity**
> To reduce reliance on LLM judges, we replace the admission function with a semantics-aware rule inspired by semantic entropy [1].
>
> We use bidirectional entailment: $\hat{y}$ is admitted if it mutually entails the reference $y$ given context, using a natural language inference (NLI) model [1]. Concretely, we check whether each report logically implies the other given the initial state, goal, and environment. Admission is granted only if both directions exceed a threshold, ensuring the two reports describe the same trajectory outcome, constraints, and success/failure status, even if phrased differently.
>
> **5) Computational cost and scaling**
> The cost has three parts.
> (i) Offline: calibration and model fitting, done once; scales linearly with data.
> (ii) Per-query: generating counterfactuals via simulation. If one rollout costs $c_{\mathrm{sim}}$ and $N_\lambda$ rollouts are used, cost is $N_\lambda \cdot c_{\mathrm{sim}}$.
> (iii) Guarantees: smaller target risk or higher confidence require larger $N_\lambda$.
>
>
> We will include a more detailed discussion of this in the Appendix of the paper.
>
> ---
>
> **References**
>
> [1] Kuhn, L. et al., “Semantic Uncertainty: Linguistic Invariances for Uncertainty Estimation in Natural Language Generation”, 2023.
>
> [2] Choi, J. et al., “Reachability-based Temporal Logic Verification for Reliable LLM-guided Human-Autonomy Teaming”, 2026.

---

> > ### Author Rebuttal · Reviewer_pzr9 · 2026-04-05
> >
> > The rebuttal addresses my main concerns reasonably well.
> >
> > The additional experiment beyond the original wireless-control setting is a useful step toward supporting the broader claims, and the simulator-fidelity analysis helps clarify how the method behaves under model mismatch. The clarification of the conformal guarantee—particularly its dependence on the admission function—is also helpful.
> >
> > I also appreciate the move beyond a single LLM-judge-based evaluation, although I still think the assessment of semantic fidelity is somewhat fragile in general.
> >
> > That said, I still see the contribution as largely integrative, and the broader generality claims should be framed with a bit more care. Overall, though, the authors have responded constructively and the paper is stronger for it.

---

### Official Review · Reviewer_kSNF · 2026-03-09

**Soundness:** 3
**Presentation:** 3
**Significance:** 2
**Originality:** 2
**Overall Recommendation:** 4
**Confidence:** 3

**Summary:**

This paper proposes a framework that enables an LLM-based agent to perform counter factual reasoning. Specifically, the authors model the interaction as SCM and propose a framework called CCG to ensure statistical reliability of the inferences provided by the LLM-based agent.

**Compliance With Llm Reviewing Policy:**

Affirmed.

**Final Justification:**

The authors have addressed my questions. As such I have raised my score by 1.

**Key Questions For Authors:**

1. Although the authors provide examples of prompts in Appendix C, it would be helpful to provide at least one specific example prompt for each method to help us determine whether LLM-as-a-judge is working well. Specifically, I'd like to know if you could provide specific examples of prompts, and examples that demonstrate that LLM-as-a-judge's judgment is reasonable.
2. It seems the same gumbel-max distribution was used for a report generation, but are the parameters used here the same as action generator? These values ​​need to be specifically described in the paper.
3. The paper conducted experiments using a very plausible simulator. However, building a realistic simulator for applications other than those presented in the paper is extremely difficult. Does this mean the methodology proposed in this paper is difficult to apply to general situations?
4. The author claims that SCM is a one of their core contributions, but I believe it's merely a diagram. Could the author elaborate on why the author considers it a core contribution?

**Limitations:**

Yes

**Strengths And Weaknesses:**

Soundness
- Using the framework CCG they propose, they appropriately demonstrated counter factual reasoning for communication relay as an example.

Presentation
- There's no Preliminary section, unless you're a complete expert in this field, this article isn't self-contained, making it difficult to follow. For example, the concept of counterfactual and why it's important should be explained in detail in a Preliminary section.
It's a minor issue, but in Figure 3, probability is denoted as D, and later as P. I think it would be better to integrate this.
In the text, "action generator" and "report generator" are highlighted in bold.
While the figure suggests a possible inference, it's important to clearly indicate which parts are "action generator" and "report generator" to maintain consistency with the text.
There are many expressions like "in (author)" in the body of the paper, so it would be better to change it to "in author."


Significance
- I think it is meaningful to show counterfactual reasoning under LLM-based environmental interaction. However, even with a statistically significant Counterfactual answer, the effectiveness of the proposed method in real-world cases needs to be fully described. Applying the proposed method to a real-world system with only statistically significant values ​​is unlikely to yield any practical operational benefits.
- The method proposed in this paper is believed to require a simulator that closely mimics the real world. Obtaining such a simulator is considered difficult, and in that respect, I believe this paper has limitations.

Originality
- It was diagrammed as SCM, including interactions with the environment.
The authors introduced CCG and showed that it was statistically valid.
Although the authors did express SCM as their contribution, it is considered a trivial contribution, as it merely diagrams what is intuitively understandable.

---

> ### Author Rebuttal · Authors · 2026-03-27
>
> We thank the reviewer for the thoughtful feedback. We address the main concerns below.
>
> **1) Presentation and clarity**
> We currently introduce the concept of conuterfactuals through the example in the introduction. We agree that a short preliminaries section would improve accessibility. In the revision, we will consider adding a concise background on SCMs, counterfactuals, and conformal calibration. We will also fix the notation inconsistency in Fig. 3 and clarify the roles of the action and report generators.
>
> **2) Practical operational benefits**
> The goal of CCG is not merely statistical significance, but reliable decision support. In many systems, e.g., robotics, operators must evaluate “what-if” scenarios before or after execution. Our framework provides set-valued counterfactual predictions with guarantees, enabling feasibility assessment. For example, if no admissible counterfactual satisfies a safety constraint, the system can flag the intent as infeasible. Thus, the value lies in risk-aware decision making rather than point prediction.
>
> **3) Requirement of a simulator**
> We agree simulator quality matters, but emphasize that our method targets domains where simulators or digital twins are already standard such as telecommunications and robotics [1–3]. Our contribution is a statistically principled layer enabling reliable counterfactual reasoning on top of such models.
>
> The method does not require a perfect simulator. As shown in our ablations and a new experiment (see response to Reviewer pzr9), performance degrades gracefully with fidelity: lower quality affects tightness, not validity. Approximate or learned simulators (e.g., world models [9,10]) can also be used, with guarantees relative to the chosen model. We will clarify this scope.
>
> **4) Role of SCMs**
> We respectfully disagree that the SCM is merely diagrammatic. It is the formal model that defines counterfactuals via abduction and intervention [4,5].
>
> In our setting, the system is a closed-loop pipeline: prompt → actions → stochastic environment → trajectory → report. In this pipeline, the counterfactual report cannot be meaningfully specified without an SCM that fixes the underlying randomness.
>
> While prior work uses SCMs to model components of the LLM [6], our contribution is to extend this to the full agent–environment loop, including explicit modelling of environment randomness and shared latent noise across counterfactual rollouts. This enables principled counterfactual reasoning over trajectories and reports.
>
> **5) LLM-as-a-judge and examples**
> We agree that adding examples would improve transparency and will include them. For instance:
> - Ground truth: “Reached target and avoided all restricted zones.”
>   Candidate: “Navigated close to the target without entering restricted areas.” → Match
> - Ground truth: “Reached target but violated a restricted zone.”
>   Candidate: “Safely avoided all restricted areas.” → No match
>
> Manual inspection of 30 cases shows strong agreement with human judgment. Importantly, CCG is not tied to LLM judges: alternative admission rules, e.g., bidirectional entailment [7] (see response to Reviewer pzr9), ROUGE, and BERTScore (see Sec. 4.1) can also be used.
>
> **6) Gumbel–Max noise and SCM interpretation**
> The action generator and report generator are modeled as distinct stochastic mechanisms in the SCM, and therefore use separate and independent exogenous noise variables. Both follow the same Gumbel–Max decoding principle, but correspond to different structural equations (see Sec. 3.1). We will further clarify this modeling choice in Sec. 3.1 and Fig. 3.
>
> **7) Applicability**
> We agree high-fidelity simulators are not always available, but our method is a modular layer: given any model supporting rollouts, it enables reliable counterfactual reasoning. It is most effective in domains where simulators or digital twins are already widely used (e.g., networking, control, cyber-physical systems) [1–3,8], but also extends to learned models such as world models [9,10].
>
> ---
>
> **References**
>
> [1] Ruah, C., Simeone O., “A Bayesian Framework for Digital Twin-Based Control, Monitoring, and Data Collection in Wireless Systems”, 2023.
>
> [2] Testolina P. et al., “Boston Twin: the Boston Digital Twin for Ray-Tracing in 6G Networks”, 2024.
>
> [3] Tao F. et al., “Digital twin in industry: State-of-the-art”, 2018.
>
> [4] Pearl, J., “Causality: Models, Reasoning, and Inference”, 2009.
>
> [5] Peters, J. et al., “Elements of Causal Inference”, 2017.
>
> [6] Li, Y. et al., “Counterfactual Token Generation in Large Language Models”, 2024.
>
> [7] Kuhn, L., et al., “Semantic Uncertainty: Linguistic Invariances for Uncertainty Estimation in Natural Language Generation”, 2023.
>
> [8] Parnianifard A. et al., “Digital-Twins towards Cyber-Physical Systems: A Brief Survey”, 2022.
>
> [9] Ha D. and Schmidhuber J., “World Models”, 2018.
>
> [10] Hafner D. et al., “Learning Latent Dynamics for Planning from Pixels”, 2019.

---

> > ### Author Rebuttal · Reviewer_kSNF · 2026-04-01
> >
> > I think the authors did a good job resolving the issues I raised especailly regarding simulator and applicability.  So I've raised the score.

---

### Official Review · Reviewer_WSHx · 2026-03-11

**Soundness:** 3
**Presentation:** 3
**Significance:** 3
**Originality:** 3
**Overall Recommendation:** 4
**Confidence:** 3

**Summary:**

This paper studies a simple but important question: if the user had phrased their intent differently, would the LLM agent act differently, and would the final outcome be better? The method builds an SCM over the user prompt, agent action, environment trajectory, and final report. It then uses abduction + counterfactual rollout + conformal prediction to return a set of counterfactual reports instead of just one. The main idea is interesting, and the set-valued output is a nice fit for this problem.

**Compliance With Llm Reviewing Policy:**

Affirmed.

**Key Questions For Authors:**

- What exactly is guaranteed here: correctness of the counterfactual explanation, or only coverage under the admission rule?

- Since the admission rule uses an LLM judge against a reference report, how sensitive are the results to judge errors or bias?

- What part of this method still works when no true counterfactual reference is available, which is the normal case in deployment?

- How often do paraphrases that are supposed to express the same intent actually lead to the same action under the same latent noise? This seems central to the paper’s motivation.

- Can the authors separate “different intent” from “different wording of the same intent” more clearly in both setup and evaluation? Right now these seem mixed together in the editable set.

**Limitations:**

1. The guarantee is weaker than it first sounds. What is guaranteed is coverage with respect to the admission function, not true counterfactual correctness in the environment. In the experiments, admissibility is decided by an LLM judge against a reference report. So the guarantee is only as good as that proxy.

2. The whole setup depends a lot on having a true counterfactual reference during calibration and evaluation. That is possible in simulation, but usually not available in real use. So right now the method looks more convincing as a simulation-based analysis tool than as something ready for real deployment.

3. For intent. The paper allows both “different intent” and “different wording of the same intent” inside the edited prompt set, but it does not actually verify that two different phrasings are understood as the same intent by the LLM. Prompts go directly to actions; there is no explicit intent variable being recovered first. So the method cannot really guarantee that it is studying “same intent, different expression” rather than just “different prompt, different behavior.”

**Strengths And Weaknesses:**

The problem is meaningful. The paper does not only change text; it also models how the agent action and environment outcome change together. That makes the setup much better than plain prompt editing. The method is also fairly clean. The SCM, latent-noise abduction, counterfactual rollout, and conformal wrapper are logically connected.

---

> ### Author Rebuttal · Authors · 2026-03-27
>
> We thank the reviewer for the thoughtful review and positive assessment. We address the main points below.
>
> **1) What is guaranteed?**
> We agree this should be stated more sharply. The guarantee provided by CCG is not an absolute notion of “true counterfactual correctness.” Rather, it is defined with respect to the chosen admission rule $A$. Specifically, CCG guarantees that the returned set contains at least one candidate that is admissible under rule $A$. For tasks with verifiable correctness, the admission function can directly produce an objective assessment of the validity of the output. In contrast, for free-form language generation, “correctness” must be operationalized via a task-dependent admission function comparing candidate and reference outputs, such as LLM-as-a-judge metrics. Overall, the proxy-relative nature of the guarantee is not a weakness of our method, but an inherent aspect of evaluating semantic fidelity. We will revise the abstract and Sec. 4 to make this interpretation explicit and avoid any ambiguity.
>
> **2) Sensitivity to judge error / bias**
> We agree that when $A$ is implemented via an LLM judge, the practical meaning of the guarantee inherits the judge’s notion of semantic similarity. However, CCG is not tied to LLM-as-a-judge: as discussed above, for tasks with verifiable ground-truth outputs, the admission function $A$ can directly evaluate the correctness of the counterfactuals.
>
> To address this concern more directly, we have performed a new experiment using the semantic clustering method used in [1] as the admission criterion. This approach groups outputs into equivalence classes based on bidirectional entailment (see Point 4 of response to Reviewer pzr9 for details). Empirically, we have observed consistent behavior across admission rules, indicating that our conclusions are not driven by a specific judge. For example, on 30 true–estimated counterfactual report pairs, both LLM-as-a-judge and semantic clustering fully matched human judgments.
>
>
> **3) What works without counterfactual references?**
> Only the conformal calibration step requires paired counterfactual references $(T,X',Y_{X'}(T))$. In contrast, the core pipeline—(i) SCM specification, (ii) abduction of latent noise $U_Z$ from the factual trajectory, (iii) counterfactual rollout under edited prompt $X'$ with shared randomness, and (iv) generation of candidate reports—does not require access to true counterfactuals at deployment time. Thus, CG can be used in practice without ground-truth counterfactual labels; what requires such labels is only the one-time calibration step that enables statistical guarantees.
>
> Importantly, calibration does not require online access to counterfactuals. In practice, these can be obtained offline via controlled data collection: e.g., in networked systems, operators routinely run A/B tests, shadow deployments, or digital-twin replays where the same context (traffic, topology, time of day) is evaluated under alternative actions with matched randomness. Historical logs with sufficient context can also be reused to approximate such paired outcomes. Calibration can thus be performed once using such data, after which CCG can be deployed.
>
> Hence, while our experiments use simulators for clarity, the method is not limited to simulation: it can rely on standard offline evaluation protocols already used in practice, and requires counterfactuals only for one-time calibration to enable statistical guarantees.
>
> **4) Paraphrases and intent vs. wording**
> We agree this distinction should be clearer. Our method does not assume that paraphrases preserve actions. Instead, it estimates the agent’s behavior under the edited prompt. If the agent is invariant to paraphrases, the predicted outcomes will be similar; if not, the method will aim to correctly reflect the induced differences. In this sense, paraphrase invariance is a property of the agent, not an assumption of CG.
>
> To explicitly disentangle these effects, in a new experiment (see response to Reviewer pzr9), we included paraphrasing the intent as a form of allowed edit to create counterfactual prompts. This separation allows us to distinguish “different intent” from “different expression of the same intent,” and show that CG appropriately captures both types of variation.
>
> ---
>
> **Reference**
>
> [1] Kuhn, L., Gal, Y., and Farquhar, S., “Semantic Uncertainty: Linguistic Invariances for Uncertainty Estimation in Natural Language Generation”, 2023.

---

> > ### Author Rebuttal · Reviewer_WSHx · 2026-04-03
> >
> > The rebuttal adequately addresses my main concerns. In particular, it clarifies the scope of the guarantee, explains the role of offline calibration versus deployment, and better separates prompt paraphrasing effects from intent changes. The additional evidence on admission criteria is also helpful. I encourage the authors to make these points explicit in the final version.

---

### Decision · Program_Chairs · 2026-04-30

**Decision:**

Accept (regular)

**Comment:**

This paper studies counterfactual reasoning for LLM-based agents by modeling the user–agent–environment loop as a structural causal model and introducing Conformal Counterfactual Generation (CCG) to produce set-valued counterfactual outcomes with coverage guarantees. Reviewers agreed that the problem is meaningful and timely and viewed the technical framework as coherent and well-motivated. The main strengths are the principled integration of SCM-based counterfactual reasoning with conformal calibration, the clear formulation of uncertainty through set-valued outputs, and empirical improvements over simple re-execution/interventional baselines. The paper is technically solid, addresses an important problem, and offers a useful framework that others are likely to build on.